# Splice modulators target *PMS1* to reduce somatic expansion of the Huntington's disease-associated CAG repeat

Zachariah L. McLean[1,2,3], Dadi Gao [1,2,3], Kevin Correia[1], Jennie C. L. Roy [1,2], Shota Shibata [1,2,3], Iris N. Farnum [1], Zoe Valdepenas-Mellor[1], Marina Kovalenko[1], Manasa Rapuru[1], Elisabetta Morini [1,2], Jayla Ruliera[1], Tammy Gillis[1], Diane Lucente[1], Benjamin P. Kleinstiver [4,5], Jong-Min Lee [1,2,3], Marcy E. MacDonald[1,2,3], Vanessa C. Wheeler[1,2,3], Ricardo Mouro Pinto [1,2,3] & James F. Gusella [1,3,6] ✉

Huntington's disease (HD) is a dominant neurological disorder caused by an expanded *HTT* exon 1 CAG repeat that lengthens huntingtin's polyglutamine tract. Lowering mutant huntingtin has been proposed for treating HD, but genetic modifiers implicate somatic CAG repeat expansion as the driver of onset. We find that branaplam and risdiplam, small molecule splice modulators that lower huntingtin by promoting *HTT* pseudoexon inclusion, also decrease expansion of an unstable *HTT* exon 1 CAG repeat in an engineered cell model. Targeted CRISPR-Cas9 editing shows this effect is not due to huntingtin lowering, pointing instead to pseudoexon inclusion in *PMS1*. Homozygous but not heterozygous inactivation of *PMS1* also reduces CAG repeat expansion, supporting *PMS1* as a genetic modifier of HD and a potential target for therapeutic intervention. Although splice modulation provides one strategy, genome-wide transcriptomics also emphasize consideration of cell-type specific effects and polymorphic variation at both target and off-target sites.

Huntington's disease (HD, MIM: 143100) is a dominantly inherited neurodegenerative disorder whose motor, cognitive, and behavioral manifestations are caused by an expanded CAG repeat in the first exon of *HTT*, which encodes huntingtin[1]. The inherited repeat, whose length negatively correlates with HD age-at-onset, undergoes further expansion in various somatic tissues but particularly in the brain[2,3], with the largest postmortem expansions found in those individuals with the earliest onset[4]. Human genome-wide association studies (GWAS) have revealed that HD age-at-onset is influenced by some DNA repair genes that play a role in repeat instability[5]. These features, together with the similar age-at-onset and lack of increased severity in individuals with two expanded alleles[6,7], have led to a sequential two-step model for HD pathogenesis wherein 1) the inherited CAG repeat lengthens over an individual's lifetime in cells that enable CAG repeat expansion, and 2) once the CAG repeat reaches a cell type-specific threshold length, it triggers toxicity/dysfunction that leads eventually to cell death[8,9]. The ultimate mechanism of toxicity is still unclear. Candidates include dysfunction caused by mutant huntingtin or amino-terminal fragments containing a lengthened CAG repeat-encoded polyglutamine segment[10] and toxicity via *HTT* mRNA[11,12].

[1]Molecular Neurogenetics Unit, Center for Genomic Medicine, Massachusetts General Hospital, Boston, MA 02114, USA. [2]Department of Neurology, Harvard Medical School, Boston, MA 02115, USA. [3]Medical and Population Genetics Program, the Broad Institute of M.I.T. and Harvard, Cambridge, MA 02142, USA. [4]Center for Genomic Medicine and Department of Pathology, Massachusetts General Hospital, Boston, MA 02114, USA. [5]Department of Pathology, Harvard Medical School, Boston, MA 02115, USA. [6]Department of Genetics, Blavatnik Institute, Harvard Medical School, Boston, MA 02115, USA. ✉ e-mail: jgusella@mgh.harvard.edu

The two-step mechanism proposed to explain HD pathogenesis also suggests two distinct therapeutic options, one to prevent CAG repeat expansion by early intervention and the other to reduce the toxicity process initiated by the somatically expanded CAG repeat. More translational attention has been paid to the toxicity step, where attempts to reduce *HTT* mRNA/protein level by targeted genetic approaches have included antisense oligonucleotides (ASOs) and RNA interference, and *HTT* transcript splice modulation[13]. Branaplam (Novartis) and PTC518 (PTC Therapeutics) are small molecules that have been in phase II clinical trials for HD based on their modulation of *HTT* splicing. The chemical structure of PTC518 has not been disclosed, but PTC Therapeutics has previously reported that risdiplam, a drug used to treat spinal muscular atrophy (SMA), also targets *HTT* with lower potency[14]. These splice modulators stabilize non-canonical nGA 3'-exonic motifs, resulting in the inclusion of a frame-shifting pseudoexon between *HTT* exons 49 and 50[14,15], with consequent lowering of huntingtin level. Recently, the VIBRANT-HD clinical trial of branaplam in adults with HD (phase 2b, Novartis) was halted due to safety concerns[16], highlighting the need for further research into the effect on HD cells, including the role of off-target splice modulation.

For designer therapeutics based on genetic targets, polymorphic sequence variation can potentially affect both on- and off-target efficacy. Consequently, we explored the effects of genetic variation surrounding the *HTT* pseudoexon and predicted alternative targets in other loci in human lymphoblast cell lines (LCLs) of defined genotype. We found that the pharmacodynamic effectiveness of the splice modulators branaplam and risdiplam can be influenced by genetic variants, both at *HTT* and other genes, where they promote pseudoexon inclusion. Interestingly, these drugs also reduced the rate of *HTT* CAG expansion in an in vitro model of repeat instability. We show the splice modulators also target *PMS1*, a known modifier of HD age-at-onset, suggesting that their suppression of CAG expansion is due to pseudoexon inclusion in *PMS1*. Inactivation of *PMS1* in the cell model dramatically reduced *HTT* CAG expansion, supporting this gene or its products as potential targets for reducing somatic CAG repeat instability to treat the first stage of HD pathogenesis.

## Results

### Splice modulator-induced products and dose-response

We treated lymphoblastoid cell lines (LCLs) from HD individuals with branaplam or risdiplam to confirm splice modulation of *HTT*. In each case, two alternatively spliced products were produced. One RNA included the pseudoexon (exon 50a in Fig. 1a, b) from novel 3' and 5' splice sites (ss) between the exon 49 and 50 sequences. The other resulted in the lengthening of exon 50 (exon 50b) via the use of the same novel alternative 3'ss as the pseudoexon (Fig. 1a, b, Supplementary Fig. 1). Both alternatively spliced products are predicted to share the same functional outcome since the inclusion of these pseudoexons introduces a premature termination codon into the *HTT* transcript. The compounds produced a dose-dependent decrease in the *HTT* canonical isoform (Fig. 1c), with branaplam (IC50 25 nM) approximately 25 times more potent than risdiplam (IC50 636 nM), but they differed in the relative proportion of the two novel products. Branaplam produced a mean ratio of exon 50b to exon 50a of 2.7 across the concentration gradient, while risdiplam displayed a lower exon 50b to 50a ratio of 0.30 (Fig. 1c).

Branaplam has been shown to bind the novel exon 50a 5'ss with U1 snRNP to enable inclusion of this pseudoexon[14]. Therefore, the prominent production of a new product in which pseudoexon 50b is generated by the novel 3'ss but utilizes the canonical exon 50 5'ss was unexpected. We postulated that the relative strength of neighboring splice sites might influence the generation of the exon 50b product. We reasoned that, due to the stronger upstream exon 49 5'ss, the initial portion of intron 49 up to the pseudoexon 50a 3'ss might be spliced out first, but the intron section downstream of the pseudoexon 50a 5'ss be retained due to the relative weakness of the latter. This hypothesis predicted that weakening the upstream site would decrease the exon 50b/exon 50a ratio produced after drug treatment. Therefore, we used site-directed mutagenesis in a minigene construct to vary the final base of exon 49 from the normal GAG|gt exon-intron junction (highlighted by |) to GAC|gt, GAT|gt, and GAA|gt (mutated nucleotide underlined). When transfected into HEK293T cells and analyzed with polymerase chain reaction (PCR) specific to the minigene, without branaplam, the GAC|gt and GAA|gt mutants each

**Fig. 1 | Branaplam and risdiplam treatment of HD LCLs produced two major *HTT* alternative splice products. a** Schematic diagram showing the alternative *HTT* splice products upon drug treatment. **b** PCR from exon 49–50 showing the size of the splice products from a single experiment. **c** Branaplam and risdiplam dose response for each *HTT* splice product. **d** Quantification of splice products produced from mutant minigenes following transfection of HEK 293 T cells either treated with a vehicle control (DMSO) or 100 nM branaplam. Source data are provided as a Source Data file.

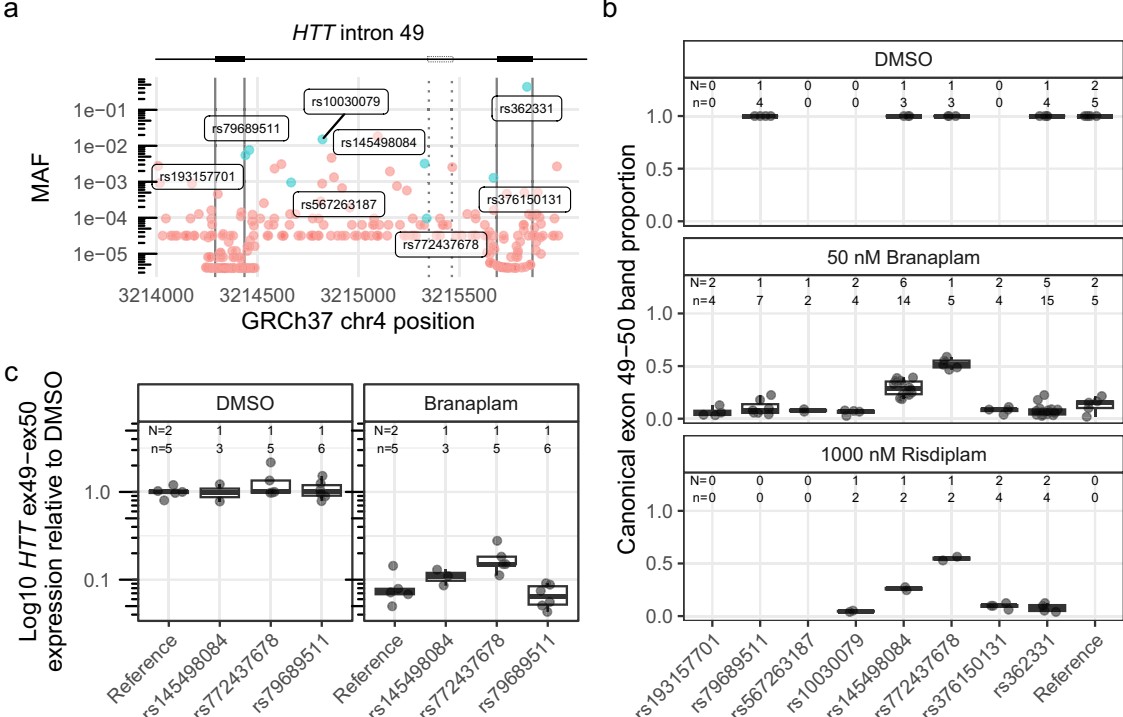

**Fig. 2 | Two single nucleotide variants affected *HTT* splice modulation. a** Minor allele frequency (MAF) of variants spanning *HTT* exon 49–50 (exons marked with solid vertical lines), with variants represented in the cell lines tested labeled and highlighted in blue. The dotted vertical lines indicate the pseudoexon splice sites (ss). **b** The proportion of canonical *HTT* exon 49–50 product across tested cells lines, grouped by heterozygous presence of variant. Since the production of the pseudoexon requires drug treatment, only a subset of the cell lines were treated with DMSO control. **c** Absolute quantification by ddPCR across exon 49–50 junction for a subset of the cell lines on a log10 axis. Box plots show the 25th and 75th percentiles (box), median (horizontal line), and range (whiskers, which are capped at 1.5x the inter-quartile range). N = Number of cell clones, n = cultures analyzed. Source data are provided as a Source Data file.

resulted in ~30% unspliced minigene product, with GAT|gt at 50% unspliced. With branaplam treatment, the ratio of exon 50b/exon 50a decreased from 4.6 for the GAG|gt minigene to 0.9 for GAC|gt, 1.1 for GAT|gt, 1.2 for GAA|gt (Fig. 1d), indicating that the relative strength of the upstream exon 49 5′ss influences branaplam-induced splicing outcomes.

**Rare sequence variants affect HTT splice modulation**

Given this evidence for sequence context impacting the effects of branaplam treatment, we evaluated the effect of genetic variation surrounding the *HTT* pseudoexon on drug-induced splice modulation. Population-based estimates from gnomAD (global ancestry) indicated that *HTT* intron 49 has low genetic variation, with no variants of minor allele frequency (MAF) > 10% and only two > 1%. We screened our bank of previously genotyped HD LCLs and identified 15 lines collectively representing eight single nucleotide variants (SNVs) of interest. We included one common variant (rs362331) located in exon 50 (Fig. 2a) and seven less frequent variants distributed across intron 49, with two close to the 5′ss (rs193157701, rs79689511), one close to the intron 49 3′ss (rs376150131), and four centrally located (rs10030079, rs145498084, rs567263187, rs772437678). Of the latter, rs772437678 and rs145498084 are located 11 and 21 nucleotides upstream of the pseudoexon 3′ss, respectively (Fig. 2a). We did not have cell lines with rs148430407, a rare SNV located 2 nucleotides downstream of the pseudoexon 5′ss that alters the canonical 5′ss intron sequence from gt to gg, primarily in individuals of African ancestry.

Treatment of the HD LCLs with 50 nM branaplam reduced the proportion of canonical splice product to 0.098 (95% confidence interval (CI): 0.021 to 0.17) in cells homozygous for the reference sequence but only to 0.49 (95% CI: 0.41 to 0.57) and 0.32 (95% CI: 0.25 0.39) ($p < 0.0001$ in both cases) in cells heterozygous for rs772437678

or rs145498084, respectively (Fig. 2b). The remaining cell lines with variants of interest showed a similar proportion of canonical splice product to those with the reference sequence ($p \geq 0.2$). The relatively higher fraction of canonical splice product remaining in cell lines with rs772437678 and rs145498084 is presumed to derive from interference by the minor allele of the respective SNV with the branaplam mechanism. We observed a similar result for these two SNVs with 1000 nM risdiplam treatment (Fig. 2b). Notably, the common SNV rs362331, located 28 bases upstream of the exon 50 5′ss, did not affect splice modulation ($p > 0.17$). Given the robust interference with splice modulation by rs772437678, we repeated the branaplam dose-response experiment with cell lines respectively heterozygous for rs772437678 or, as a control, rs79689511. At higher branaplam concentrations, the proportion of the canonical isoform continued to decrease in cell lines with rs772437678 but was consistently higher than in the control (Supplementary Fig. 2). We also analyzed the exon 50b/exon 50a ratio for this set of cell lines and observed no differences from the samples with reference sequence ($p \geq 0.1$) (data not shown).

Although the densitometric method permits comparison of the relative levels of canonical and non-canonical splice variants, we expected that the absolute level of *HTT* mRNA might be reduced by preferential degradation through nonsense-mediated mRNA decay (NMD) of the noncanonical products due to their premature termination codon. Consequently, we performed droplet digital PCR (ddPCR) for accurately quantifying the *HTT* canonical isoform, analyzing a subset of the same samples using a hydrolysis probe spanning the exon 49–50 junction. Treatment with 50 nM branaplam reduced *HTT* cDNA with the exon 49–50 junction by ~15-fold in control cells and ~7-fold in the cell line with rs772437678 (Fig. 2c), reflecting the ~2-fold relative effect seen previously by densitometry.

## An engineered cell model for investigating CAG repeat instability

Having established that DNA sequence polymorphisms can impact the effects of the splice modulators, we turned our attention to the critical driver of HD pathogenesis: *HTT* CAG repeat expansion. It has been suggested that reducing huntingtin levels by ASO treatment might also reduce CAG repeat expansion[17]. Consequently, we decided to test the effects of the *HTT*-lowering splice modulators on *HTT* CAG repeat expansion. Most cultured HD cell lines display limited CAG repeat instability, so we developed a model system for this purpose by inserting an expanded CAG repeat construct into the adeno-associated virus (AAV) integration site and safe harbor locus in intron 1 of *PPP1R12C* on chromosome 19 (AAVS1, Fig. 3a) of hTERT-RPE1 (RPE1) cells. RPE1 is a near-diploid immortalized cell line often used to study DNA repair pathways[18]. It can be arrested at G0/1 through contact inhibition by growing the cells to confluency (Fig. 3b), with the percentage of cells in S-phase decreasing from 20% to <5% within four days of confluency (Fig. 3c, Supplementary Fig. 3a). We isolated the expanded CAG *HTT* exon 1 from a juvenile-onset HD individual (115 CAGs) and knocked the fragment into the AAVS1 safe harbor locus with a doxycycline-inducible promoter, intending to control transcription and transcription-linked repeat instability. We isolated 8 clones, each with 110–115 CAG repeats, and cultured the cells in the presence and absence of doxycycline. In confluent cultures over 28 days, transgene-containing lines displayed *HTT* exon 1 CAG expansion that was greater in the non-induced state than in the presence of doxycycline (Fig. 3d, Supplementary Fig. 3b). We replicated this in a subsequent experiment in even longer cultures for 48 days (Fig. 3e, f). By day 48, confluent non-induced cultures had an average repeat gain of 7.8 (95% CI: 7.1 to 8.5), while induced cultures gained 2.6 CAGs (95% CI: 2.0 to 3.3, $p < 0.0001$). Compared to confluent cultures, dividing cultures exhibited a 1.9-fold (95% CI: 1.8 to 2.0, $p < 0.0001$) higher average repeat gain, with non-induced and induced cultures exhibiting an average repeat gain of 12.0 (95% CI: 11.4 to 12.7) and 6.8 (95% CI: 6.2 to 7.5), respectively (Fig. 3e, f).

The reduction in CAG expansion with doxycycline treatment initially seemed at odds with the dependence of repeat instability on transcription[19,20], so we assessed the expression of the inserted transgene and the *PPP1R1C* host gene. Green fluorescent protein (GFP) signal from transgene expression measured in noninduced cells was 31-fold (95% CI: 29 to 33) higher than the measurement of background autofluorescent signal in nontransgenic parental cells, indicating significant inducer-independent leaky expression. Doxycycline treatment produced a further 5.9-fold increase (95% CI: 5.8 to 5.9) (Fig. 3g). In both instances, ~20% of cells were GFP-negative, initially suggesting transgene silencing. However, upon sorting and culture of the GFP-negative population, the majority of cells were GFP-positive at 27 days, indicating that the GFP-negativity was only a temporary state (Supplementary Fig. 3c). We compared the CAG repeat instability in cell populations initially enriched by sorting for either GFP-positive or GFP-negative cells and found no difference in either uninduced or induced cultures (Supplementary Fig. 3d).

Next, we used ddPCR to quantify RNA from the inserted PuroR (puromycin resistance gene) and *HTT* exon1/enhanced GFP (EGFP) cassettes, which are transcribed in opposite directions, and from *PPP1R12C*, which is transcribed across the transgene-containing intron in the same direction as PuroR. For *PPP1R12C*, heterozygosity of RPE1 cells at rs34521018 allowed us to quantify the spliced exon1/exon2 product in an allele-specific manner, distinguishing expression of the allele with the transgene from the remaining wild-type allele. Insertion of the transgene into *PPP1R12C* intron 1 reduced the almost equal allelic ratio of 1.1 (95% CI: 1.0 to 1.2) in parental cells to 0.17 (95% CI: 0.098 to 0.24) in RPE1-AAVS1-CAG115 cells. Relative to non-induced cells, treatment with doxycycline decreased expression of *PPP1R12C* exon 1–2 from the transgene-containing allele and PuroR 1.8-fold (95% CI: 1.6 to 2.0) and 2.1-fold (95% CI: 2.0 to 2.1), respectively, while it

increased *HTT* exon 1/EGFP expression 8.2-fold (95% CI: 7.6 to 8.8) (Fig. 3h), consistent with the increase in GFP fluorescence. *HTT* exon 1/EGFP expression in non-induced cells was 3.8-fold (95% CI: 2.4 to 5.2) higher than the expression of PuroR, indicating that even in the non-induced condition, there is considerable transcription of the *HTT* exon 1/EGFP cassette over the CAG repeat. Although this level is increased by doxycycline induction, the treatment also decreases both transcription in the opposite direction and CAG repeat instability. Thus, the observation of greater CAG repeat instability in the non-induced state is consistent with the reported synergistic effect of convergent transcription on promoting CAG expansion[21,22].

We validated the relevance of our RPE1-AAVS1-CAG115 cell line to model somatic instability processes by perturbing *MSH3* and *FAN1*, modifiers of HD age-at-onset also known to influence somatic CAG repeat instability[9]. We also tested *PMS1*, another putative modifier of HD age-at-onset, whose effect on CAG repeat instability has not been previously established. We utilized CRISPR-Cas9 nuclease to target and modify the coding sequences of these genes via loss-of-function insertion or deletion mutations (indels) and analyzed repeat instability in the pooled heterogeneously-edited populations of cells (Supplementary Fig. 4a). Fragment analysis traces for the empty vector control and *FAN1*-targeting vector each showed a single approximately normally distributed population with CAG length increasing over time (Fig. 4a). As expected, *FAN1* knock-out increased the average CAG repeat gain per week from 1.34 (95% CI: 1.22–1.47) to 2.52 (95% CI: 2.40–2.64) ($p < 0.0001$) (Fig. 4b). By contrast, *MSH3* and *PMS1* knock-outs produced more complex distributions (Fig. 4a). The *MSH3* knock-out culture developed a clear bimodal CAG repeat length distribution, with one peak appearing to reflect CAG repeat contraction and the other modest, if any, expansion. The *PMS1* knock-out exhibited a small degree of expansion in some cells, albeit far less than that seen in either the empty vector or *FAN1* knock-out conditions.

To clarify the different instability distributions in *MSH3*- and *PMS1*-edited cells, we isolated clones from the pooled populations and repeated the instability analysis. For each of the genotypes, the distribution was monomodal (Supplementary Fig. 4b), suggesting that the initial distributions reflected a mixture of edited and non-edited cells that differed in their propensity for CAG expansion. From the *MSH3*-targeted population, we obtained 3 non-edited and 11 biallelically-edited clones representing complete knock-outs. The latter showed no gain in CAG repeats (−0.037 per week, 95% CI: −0.11 to 0.035) compared to a gain of 2.0 CAGs (95% CI: 1.8 to 2.1) for the non-edited lines (Fig. 4c). For *PMS1*, we derived 6 biallelically-edited and 3 monoallelically-edited clones. The biallelically-edited strains displayed a small degree of residual repeat expansion with a repeat gain of 0.13 (95% CI: 0.028 to 0.22) per week, significantly higher ($p = 0.0086$) than the equivalent *MSH3* knock-outs. While the residual instability might indicate some functional redundancy of the mismatch repair machinery partially compensating for the loss of PMS1, the *PMS1* genome editing was in exon 6, which can be alternatively spliced, so the residual repeat expansion might be due to the low-level expression of a minor isoform in RPE1 cells (Supplementary Fig. 8a). The heterozygous *PMS1*-edited lines did not differ ($p = 0.63$) from non-edited cells with a repeat gain of 1.9 (95% CI: 1.7 to 2.1) per week, indicating that a single copy of *PMS1* is sufficient to support the rate of CAG expansion observed in wild-type cells. Interestingly, western blot analysis of PMS1 confirmed the absence of intact PMS1 in the biallelically-edited lines but showed similar amounts of the protein in the wild-type and heterozygous lines (Supplementary Fig. 4c), suggesting that a single copy of *PMS1* is sufficient to allow regulation of the protein at the wild-type level, perhaps determined by association with its binding partner MLH1 in the MutL-beta complex[23,24].

Overall, our results are consistent with the view that *FAN1* and *MSH3* act as genetic modifiers of HD through their effects on somatic CAG repeat expansion and provide evidence that functional *PMS1*, like

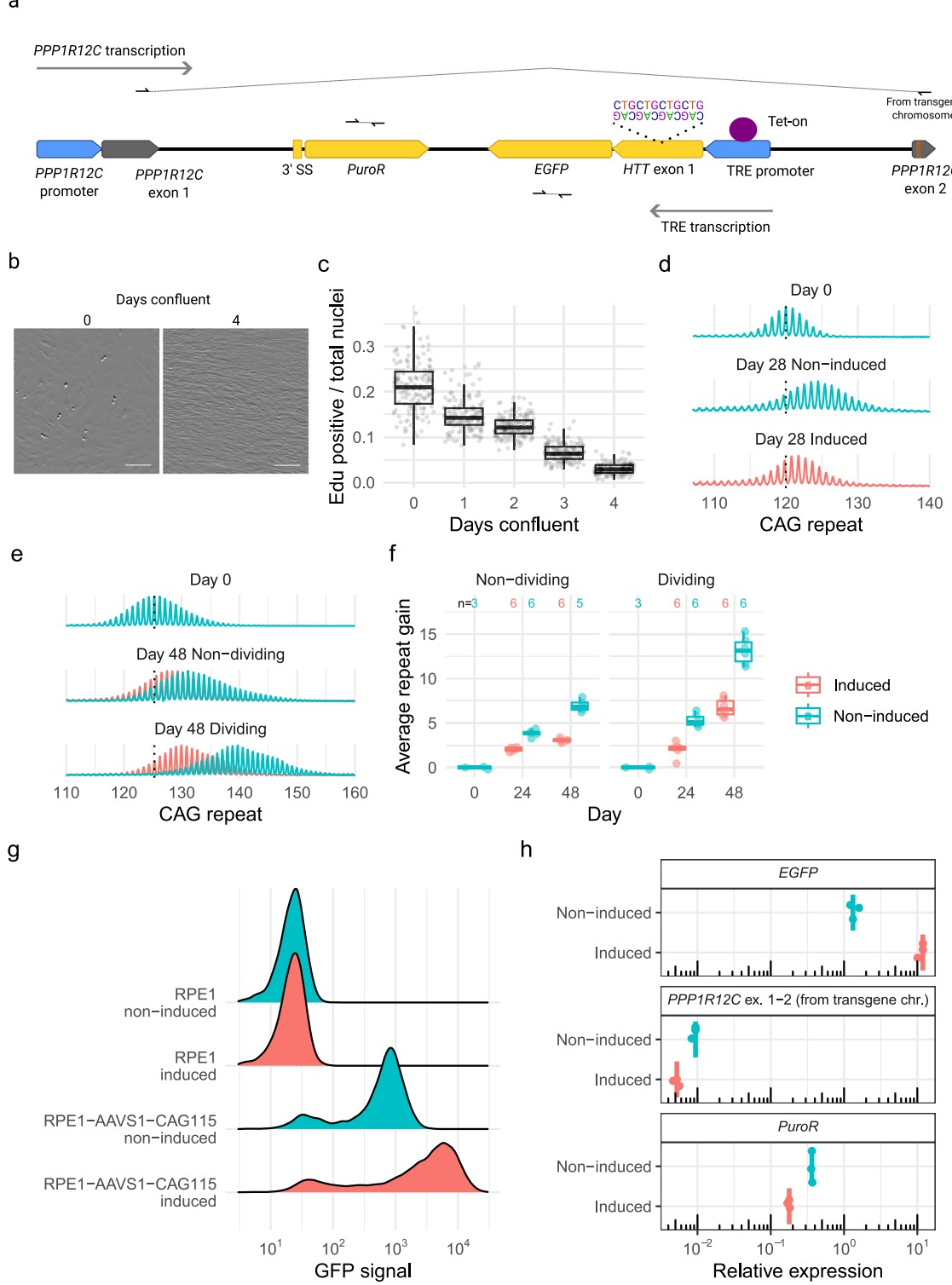

*MSH3*, supports CAG expansion. Furthermore, the rapid CAG expansion in this tractable system makes the RPE1-AAVS1-CAG115 a useful model for functional genomic investigations of CAG repeat instability.

**Branaplam and risdiplam suppress CAG repeat expansion**
To determine whether the splice modulators that reduce huntingtin expression affect CAG repeat expansion, we treated a RPE1-AAVS1-CAG115 clone maintained at confluency without doxycycline with high or low dosages of branaplam or risdiplam. Branaplam caused a dose-dependent reduction in repeat expansion, with an average CAG gain

per week of 0.94 (CI:0.88 to 1.01) in the dimethyl sulfoxide (DMSO) vehicle control, 0.81 (CI:0.75 to 0.88) at 25 nM branaplam (*p* = 0.005) and 0.73 (CI:0.66 to 0.79) at 100 nM branaplam (*p* < 0.0001) (Fig. 5a). By contrast, 100 nM risdiplam produced relatively little change in CAG repeat gain, at 0.87 (CI:0.81 to 0.93) per week compared to the control's 0.77 (CI:0.71 to 0.83) (*p* = 0.03). However, 500 nM risdiplam caused a significant decrease in the rate of repeat expansion to 0.40 (CI:0.34 to 0.46) CAGs per week (*p* < 0.0001) (Fig. 5b).

Increasing drug concentrations are expected to be associated with progressively more potent effects at both target and off-target

**Fig. 3 | RPE1-AAVS1-CAG115 cell model for CAG repeat instability. a** Schematic diagram of the key elements of transgene integration in the AAVS1 site in intron 1 of *PPP1R12C*. The *PPP1R12C* promoter (blue) drives transcription from exon 1 to 2 of the endogenous gene (gray), which was analyzed with allele specific primers due to a polymorphism in exon 2 (orange line). The puromycin resistance gene (PuroR, yellow) is driven from the same promoter with a new 3'splice site (3'SS, yellow). *HTT* exon 1 / EGFP expression (both in yellow with location of the CAG repeat shown as a partial sequence) is driven in the opposite direction from tetracycline-responsive element (TRE) promoter (blue). **b** RPE1-AAVS1-CAG115 brightfield images of cells nearing confluency (Day 0) and contact inhibited at confluency for four days (Day 4) (scale bar 0.1 mm). **c** Quantification of the number of RPE1-AAVS1-CAG115 cells in S-phase by image analysis. Edu-positive nuclei represent cells that incorporated Edu into their DNA and stained positive upon labelling, which was normalized on the total number of nuclei per field of view (160 images analyzed per group). Box plots show the 25th and 75th percentiles (box), median (horizontal line), and range (whiskers, which are capped at 1.5x the inter-quartile range). **d** CAG repeat fragment

distribution for a single RPE1-AAVS1-CAG115 clone in the absence (blue, middle) or presence (red, bottom) of doxycycline-induced transcription compared to day 0 (top). (**e**) CAG repeat fragment distribution clone in the absence (blue) or presence (red) of doxycycline-induced transcription for non-dividing (middle) and dividing (bottom), compared to day 0 (top). **f** Average repeat gain in non-dividing or dividing cells in a single RPE1-AAVS1-CAG115 clone with either non-induced (blue) or induced (red) transcription. The n indicates the number of cultures analyzed. **g** GFP fluorescence signal analyzed by flow cytometry in parental RPE1 (no transgene) and RPE1-AAVS1-CAG115 cells with either non-induced (blue) or induced (red) transcription. **h** Relative expression, normalized to expression of reference gene *SDHA*, of *PPP1R12C* exon 1–2 (allele specific from the transgene chromosome), PuroR (specific to transgene with assay designed to be unable to detect puromycin in parental RPE1 from hTERT transgene) and EGFP by ddPCR, with either non-induced (blue) or induced (red) transcription. Three replicates analyzed for each condition and data displayed on a log10 transformed axis. Source data are provided as a Source Data file.

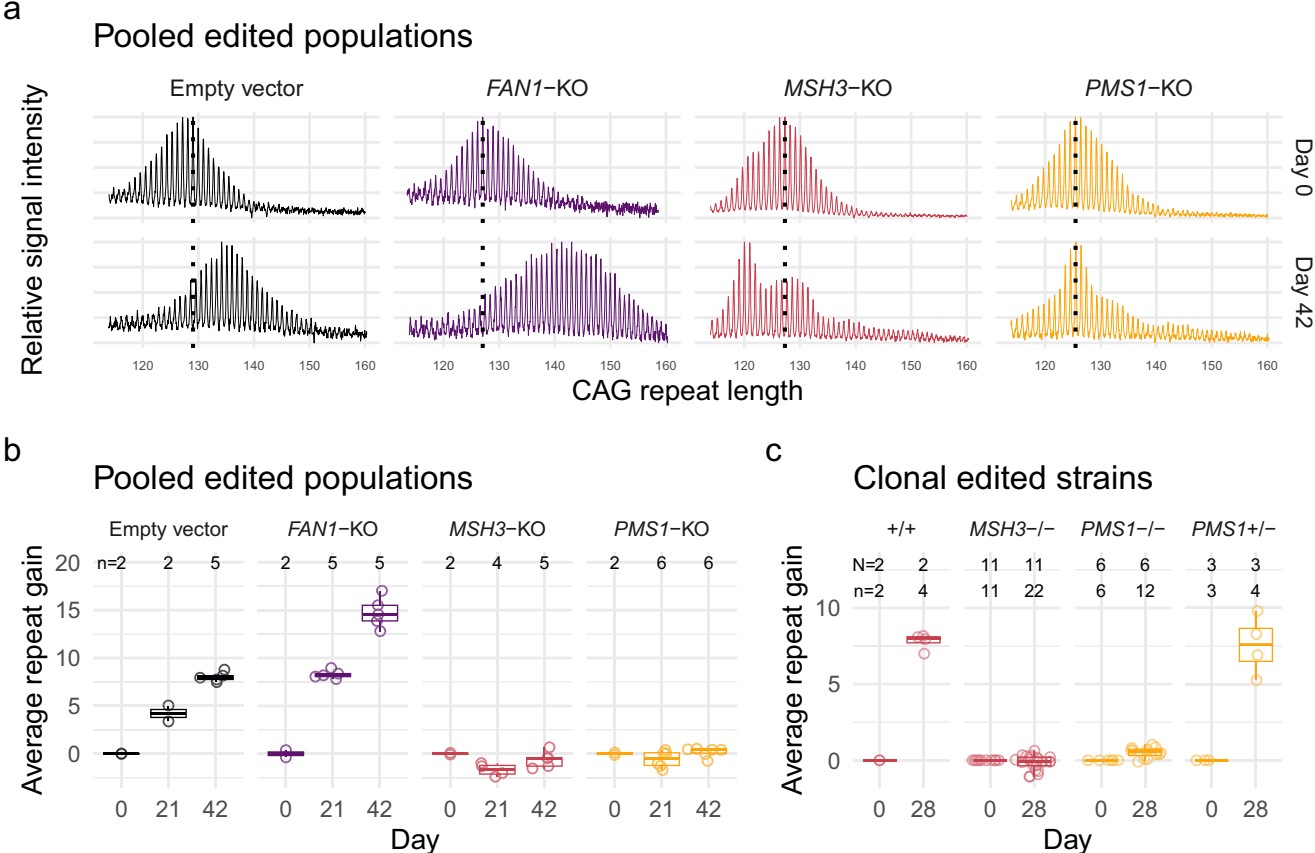

**Fig. 4 | RPE1-AAVS1-CAG115 cell model repeat expansion depends on known modifiers of HD onset and CAG repeat instability. a** Fragment analysis traces showing the change in CAG repeat length distribution across time in different non-edited and edited cells for pooled edited populations. Color indicates CRISPR-Cas9 knock-out (KO): non-targeting empty vector (black), *FAN1* (purple), *MSH3* (red), and *PMS1* (orange). The plots represent raw fluorescent signal without baseline correction and therefore have a negative signal bias with increasing fragment size. The

following instability metrics were derived from data processed in the GeneMapper software which corrects this bias. **b** Average repeat gain for pooled edited populations, with each dot representing a biological replicate. **c** Average repeat gain for cell clones isolated from either *MSH3* (red) or *PMS1* (orange) targeted populations. Box plots show the 25th and 75th percentiles (box), median (horizontal line), and range (whiskers, which are capped at 1.5x the inter-quartile range). N = Number of cell clones, n = cultures analyzed. Source data are provided as a Source Data file.

sites, so we assessed whether the reduction in repeat expansion might also be associated with increasing drug cytotoxicity. From high-throughput image analysis assays in cultures of dividing cells, proliferation was reduced in a dose-dependent manner, beginning at 250 nM branaplam and 500 nM risdiplam (Supplementary Fig. 5a). In the same experiment, acute cytotoxicity assessed by DNA labeling of dead cells showed a dose-dependent increase starting at 500 nM

branaplam, but no increase for risdiplam up to 2000 nM (Supplementary Fig. 5b). To investigate whether the drugs caused cell death longer-term in non-dividing cultures, we maintained the cells at confluency for two weeks. Compared to DMSO treatment, we observed a 3-fold increase in DNA labeling of dead cells for 200 nM branaplam ($p < 0.001$) and a 23-fold increase for 500 nM risdiplam ($p < 0.001$) (Fig. 5c). We also observed a rise in the background fluorescence in the

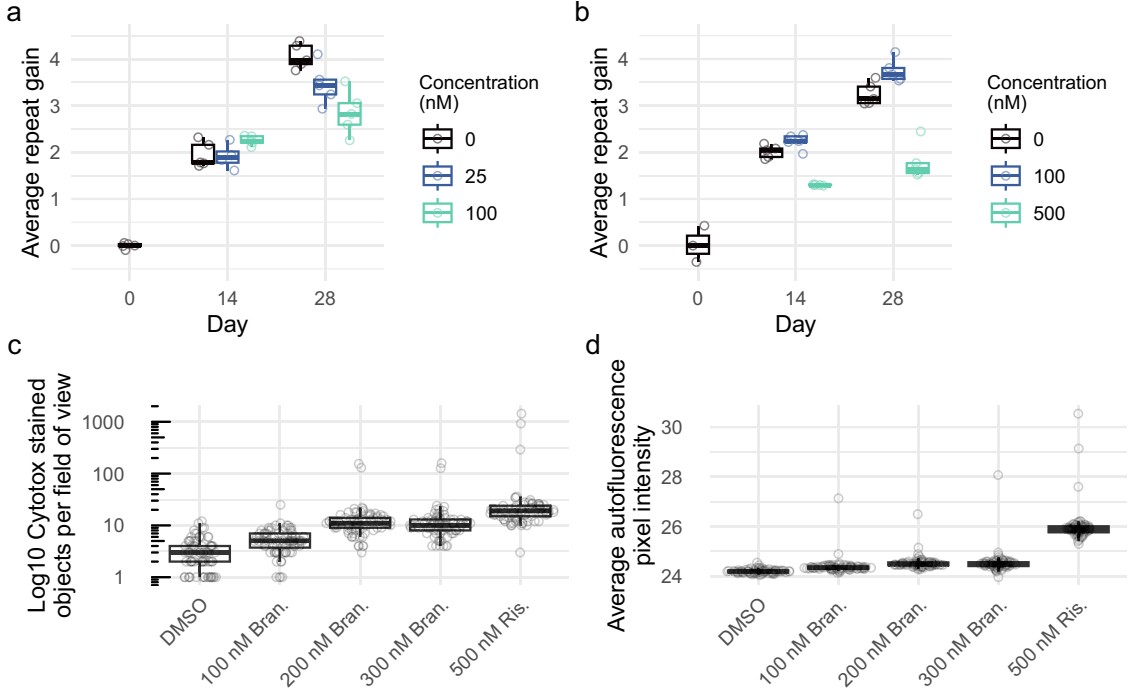

**Fig. 5 | Branaplam and risdiplam treatments reduced repeat expansion in RPE1-AAVS1-CAG115 cells.** Average repeat gain of non-induced RPE1-AAVS1-CAG115 cells with treatment of either branaplam (**a**) or risdiplam (**b**), with the color indicating the drug concentration. Each treatment group and timepoint had five cultures analyzed, except risdiplam day 0 which had three. **c** Drug cytotoxicity quantified by high-throughput image analysis of cells treated with DNA labeling of dead cells. **d** Average background autofluorescence pixel intensity. For c and d, 81 images were analyzed per treatment. Box plots show the 25th and 75th percentiles (box), median (horizontal line), and range (whiskers, which are capped at 1.5x the inter-quartile range). Source data are provided as a Source Data file.

500 nM risdiplam group (Fig. 5d, Supplementary Fig. 6), suggesting drug-induced cellular stress, which has previously been correlated with an increase in autofluorescence[25]. Thus, the effects on CAG instability at the highest drug doses are accompanied by coincident cytotoxicity, potentially due to increasing off-target effects on splicing at loci across the genome and their secondary consequences.

## HD genetic modifier PMS1 contains a drug-inducible pseudoexon

We postulated that even for low-dose branaplam, the suppression of *HTT* CAG repeat instability was likely an indirect consequence of its splice modulation, either at *HTT* or potentially at another locus. Therefore, we analyzed the list of genes with branaplam- and risdiplam-induced pseudoexons described in the RNA sequencing (RNAseq) results of previously published datasets (Supplementary data 1). Two, *PMS1* and *DHFR*, are within haplotypes associated with genetic modification of HD age-at-onset[5]. The haplotype at *DHFR* also contains the adjacent *MSH3*, a known modifier of repeat instability, but RNAseq data from Bhattacharyya et al. (Supplementary Data 1) showed that branaplam treatment significantly reduced *DHFR* mRNA but not *MSH3* mRNA[14]. Consequently, we focused on huntingtin and *PMS1* as potential mediators of the splice modulators' effects on repeat expansion.

*PMS1* contains a pseudoexon located within the 26 kb or 34 kb intron 5 (Fig. 6a), depending on the isoform (Supplementary Fig. 7). In LCLs and RPE1 cells, the predominant isoform a includes exon 6. With drug treatment, the pseudoexon is spliced into mRNA for both isoform a and the minor isoform b (Supplementary Fig. 8a). The 91 bp pseudoexon contains a stop codon (Supplementary Fig. 8b), predicted to result in a truncated PMS1 lacking the crucial C-terminal MLH1 dimerization domain and potentially trigger nonsense-mediated decay[26]. The drug-binding motif in the exon upstream of the 5′ss differs from the *HTT* pseudoexon, with AAUGA at *PMS1* compared to GCAGA at *HTT*, but both have the same downstream intronic guaag motif. Branaplam was more effective for causing *PMS1* pseudoexon inclusion in LCLs, with an IC50 of 100 nM compared to 205 nM for risdiplam (Fig. 6b). In RPE1 wild-type cells, both drugs reduced the level of PMS1 protein in a dose-dependent manner (Fig. 6c, Supplementary Fig. 8c). The drugs differ in their relative effects on *HTT* and *PMS1* pseudoexon inclusion: branaplam can preferentially target *HTT* (~4-fold higher IC50 for *HTT* over *PMS1*), while risdiplam preferentially targets *PMS1* (~3-fold higher IC50 for *PMS1* over *HTT*).

## Splice modulators reduce CAG expansion via PMS1 pseudoexon inclusion

We edited the pseudoexon locations in these two genes to determine whether pseudoexon inclusion at *HTT* or *PMS1* contributed to reducing *HTT* CAG repeat expansion. Using single guide RNAs (gRNAs) directly targeting the GA 3′-exonic motif (Fig. 7a, left) at the *HTT* pseudoexon 5′ss, we efficiently generated indels (Supplementary Fig. 9). Edited clones had an A insertion between the GA 3′-exonic motif and the GT 5′-intronic motif (Supplementary Fig. 10a). In a comparable strategy, attempts with two different gRNAs for the *PMS1* pseudoexon yielded very inefficient editing directly at the site (Supplementary Fig. 9). Therefore, we modified *PMS1* with an alternative strategy to delete a 137 bp region from the pseudoexon into the adjacent intron using dual gRNAs (Fig. 7a, right). Of the 33 clones isolated, 12 had a heterozygous deletion (representative examples in Supplementary Fig. 10b), but none was biallelically edited.

We treated representative *HTT*- and *PMS1*-edited lines with the splice modulators to determine the effect of the genome editing on both canonical and drug-induced splicing. In *HTT*, the A insertion disrupted the drug-induced pseudoexon inclusion, resulting in only canonical splicing from *HTT* exon 49–50 (Fig. 7b) despite treatment with 100 nM branaplam or 500 nM risdiplam. In cells heterozygous for *PMS1* pseudoexon editing, these treatments markedly increased the

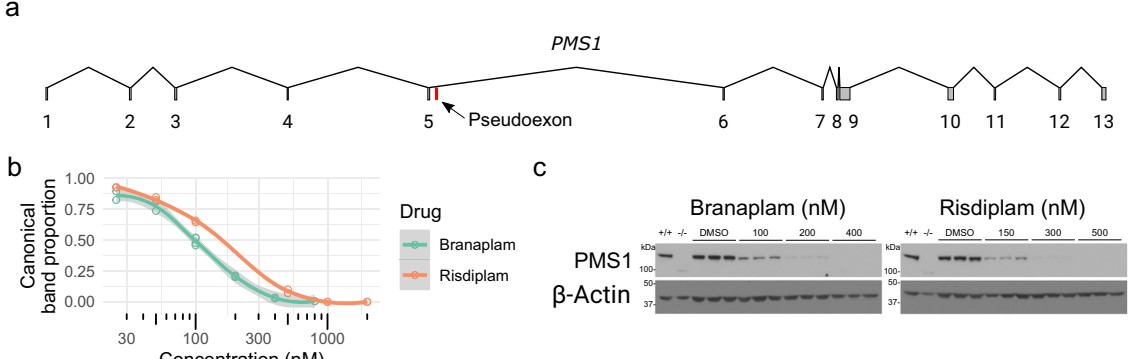

**Fig. 6 | HD modifier *PMS1* contains a drug-inducible pseudoexon. a** Schematic diagram of the *PMS1* transcript (NM_000534) highlighting the pseudoexon location in red. **b** Dose response of *PMS1* exon 5–6 (canonical band) after branaplam (teal) or risdiplam (red) treatment with each empty dot representing a biological replicate, the line showing the local polynomial regression, and the ribbon displaying the standard error. **c** Western blot showing decreased PMS1 protein in RPE1-AAVS1-

CAG115 cells after branaplam or risdiplam treatment for 10 days with DMSO or the indicated drug concentrations predicted from the RNA dose response to represent IC50, IC75, and IC90. The +/+ and -/- lanes represent non-edited wild-type and edited *PMS1* knock-out RPE1 cells, respectively. The samples are from a single experiment with three biologically independent samples per treatment. Source data are provided as a Source Data file.

proportion of canonical *PMS1* splice product (Fig. 7b). Accurate quantification of the *PMS1* canonical isoform by ddPCR showed that the *PMS1* monoallelic editing did not change the level of splicing across the *PMS1* exon 5–6 junction in the absence of drug ($p = 0.7$ relative to wild-type) (Fig. 7c). However, 200 nM branaplam treatment elicited a 3.8-fold (95% CI: 2.5 to 7.5, $p < 0.0001$) reduction in wild-type cells but only a 1.7 fold (95% CI: 1.4 to 2.1, $p < 0.0001$) reduction in the *PMS1*-edited cells ($p = 0.015$) (Fig. 7c). Thus, disrupting the sequences required for *PMS1* pseudoexon inclusion from one allele reduced the effectiveness of the splice modulators but did not affect canonical splicing.

We next quantified the repeat instability in these cell lines in 4–5 week experiments with various drug treatments. There were systematic clonal differences in the rate of repeat expansion (Supplementary Fig. 11a, b), so we normalized the data to the repeat expansion in the DMSO group for each clone. We treated the cell lines with either 100 nM branaplam for relatively stronger *HTT* splice modulation or 500 nM risdiplam for relatively stronger *PMS1* splice modulation. The removal of the *HTT* pseudoexon had no effect on repeat expansion for either 100 nM branaplam or 500 nM risdiplam (Fig. 7d), ruling out the drugs' effects on *HTT* pseudoexon inclusion as the cause of reduced CAG repeat expansion. The failure of lowering *HTT* expression to reduce CAG repeat instability in our cell model differs from the antisense-oligonucleotide (ASO) huntingtin knock-down of Coffey et al.[17] but agrees with the lack of effect on CAG repeat instability of reducing huntingtin by di-valent anti-*HTT* siRNA[27], suggesting the possibility that the ASO-treatment caused reduced *HTT* transcription, with consequent reduction of CAG instability[28–30]. Alternatively, the difference might reflect the importance of cellular context (liver and striatum in vivo or medium spiny neurons in vitro versus cultured RPE1 cells).

In contrast with *HTT*, we were unable to obtain a line homozygous for the removal of the *PMS1* pseudoexon, but this did not prevent a test of the role of splice modulator-induced pseudoexon inclusion. In the heterozygous lines, the removal of the *PMS1* pseudoexon from one allele makes that edited allele refractory to pseudoexon inclusion, while the wild-type allele remains susceptible to the effect. Notably, our previous editing experiment to create *PMS1* knock-outs showed that a single active *PMS1* allele is sufficient to support CAG instability comparable to wild-type. Consequently, splice modulator treatment of the lines with heterozygous removal of the pseudoexon presented two possibilities: 1) if *PMS1* pseudoexon inclusion plays no role in reducing CAG repeat instability, then splice modulator treatment of these lines

should show reduced CAG expansion comparable to treated wild-type cells versus 2) if *PMS1* pseudoexon inclusion contributes to reducing CAG repeat instability, then splice modulator treatment should result in a restoration of CAG expansion comparable to untreated wild-type or heterozygous PMS1-knockout cells. The results of treating the cell lines with the splice modulators were consistent with the second alternative, supporting a contribution of pseudoexon inclusion to the reduced CAG expansion observed in treated wild-type cells. They also hinted at potential additional effects of the splice modulators beyond those accounted for by *PMS1*. Indeed, treatment with 100 nM branaplam did not reduce CAG expansion of pseudoexon-edited lines compared to unedited lines but instead resulted in weak evidence of a small 1.1-fold (95% CI: 0.99 to 1.2, $p = 0.019$) increase in repeat gain. We repeated the experiment with increasing doses of branaplam to confirm the decrease in repeat expansion in wild-type cells and the slightly enhanced repeat expansion in the *PMS1* pseudoexon-edited cells. In wild-type cells, we again observed a dose-dependent effect of branaplam on reducing CAG repeat expansion, which decreased 1.2-fold at 100 nM branaplam (95% CI: 1.1 to 1.2, $p < 0.0001$), 1.5-fold at 200 nM (95% CI: 1.4 to 1.6, $p < 0.0001$) and 1.9-fold at 300 nM (95% CI: 1.7 to 2.1, $p < 0.0001$) relative to DMSO (Fig. 7e). By contrast, the *PMS1* pseudoexon-edited cells displayed repeat expansion increased by 1.2-fold at 100 nM (95% CI: 1.1 to 1.3, $p < 0.0001$) and 200 nM (95% CI: 1.1 to 1.3, $p < 0.0001$), but not at 300 nM, where treated cells appeared similar to DMSO ($p = 0.62$) (Fig. 7e). Treatment of the *PMS1* pseudoexon edited cells with 500 nM risdiplam reduced CAG expansion only 1.7-fold (95 CI: 1.4 to 2.0, $p < 0.0001$) compared to DMSO, far less than the 4.6-fold reduction (95% CI: 3.5 to 6.7, p < 0.0001) it elicited in wild-type cells (Fig. 7d), indicating that pseudoexon inclusion at *PMS1* makes a substantial contribution to risdiplam's inhibition of CAG expansion at high dosage but has a residual impact not affected by the pseudoexon editing. Overall, the results of targeting *PMS1* via the drug inducible pseudoexon explained the reduction in rate of CAG repeat expansion caused by branaplam but only partially explained the observed effect with risdiplam. The partial effect with risdiplam and the small increases in expansion with the lower branaplam doses suggest that the drugs may also have effects on splicing in other genes that can influence CAG repeat instability.

To seek clues to potential sources of these additional effects on CAG repeat instability, we examined the multiple previous reports of transcriptional effects of the splice modulators in multiple studies with different cell types: Monteys et al.[31] (HEK293, 25 nM branaplam); Bhattacharyya et al.[14], (SH-SY5Y cells, 100 nM branaplam & 500 nM

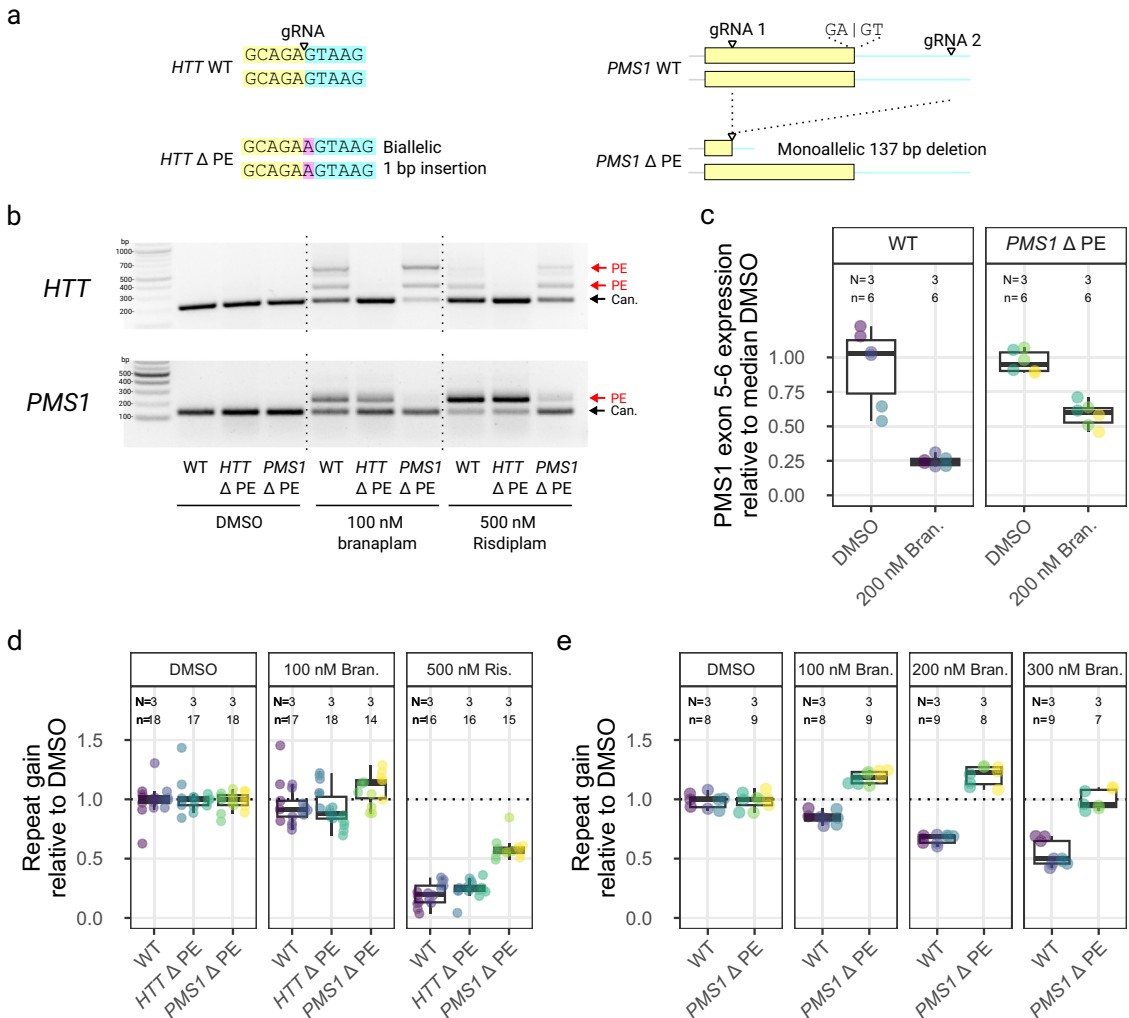

**Fig. 7 | *PMS1* pseudoexon inclusion explained the effect on CAG repeat expansion with branaplam, but only partially with risdiplam. a** Schematic diagrams showing the CRISPR-Cas9 targeting approach for the disruption of pseudoexon (PE) sequences in *HTT* (left) and *PMS1* (right). Yellow indicates pseudoexon sequence upstream of the wild-type (WT) 5′ splice site targeted by the drug, blue representing the downstream intronic sequence, with the PE mutation (ΔPE) sequence highlighted in purple. **b** PCR analysis over the *HTT* (top) and *PMS1* (bottom) pseudoexon splice junctions with branaplam or risdiplam treatment for the control and pseudoexon edited cell lines. The data are from a single

experiment. **c** Accurate quantification of *PMS1* canonical isoform by ddPCR for the control and *PMS1* pseudoexon edited cell lines. Each unique cell line is represented by a different dot color. **d, e** The average repeat gain per week after branaplam or risdiplam treatment for the different edited cell lines (dot color), normalized on the average repeat gain in the DMSO for each genotype. Box plots show the 25th and 75th percentiles (box), median (horizontal line), and range (whiskers, which are capped at 1.5x the inter-quartile range). Source data are provided as a Source Data file.

SMN-C3 (a close analogue of risdiplam)); Keller et al.[15], (SH-SY5Y cells, 100 nM branaplam); Ottesen et al.[32], (fibroblasts, 40 nM branaplam & 1000 nM risdiplam). Our survey indicated relatively low overlap between the genes identified to have drug-induced splicing alterations between the datasets (Fig. 8a). While methodological differences might account for some of the differences, these data suggest that, like normal alternative splicing, cellular context is critically important to splice modulation. In the most recent study, treatment of fibroblast with 1000 nM risdiplam, resulted in altered expression of more than 10,000 genes, including potential drug-induced splicing modifications to genes relevant to CAG repeat instability and/or modification of HD: *FAN1*, *MLH3*, *MLH1*, *TCERG1*. However, these suggested splicing modifications were all towards increased normal canonical splicing (i.e., reductions of either intron retention or exon skipping), and the change in gene expression was <1.5-fold. This study also reported a decrease in expression of the HD-modifier *LIG1*, but no significant effect on *LIG1* splicing, suggesting the reduction was an indirect secondary effect of altered splicing at another locus. Together, these data do not point to

an obvious candidate for the additional impact of risdiplam beyond its targeting of *PMS1* in our cell system. Still, they emphasize the complex considerations imposed by tissue-specific effects in developing splice modulators for clinical use.

## SpliceAI predictions on drug-responsive exons genome-wide

Given that it was an off-target effect of the splice modulators that revealed *PMS1* as an alternative therapeutic target and that sequence variants near the original *HTT* target splice site altered the drugs' effects, any use of splice modulators as potential therapeutics should consider the potential impact of polymorphic variation in the human population. Consequently, we used the deep neural network tool SpliceAI[33] to predict variants that might modulate branaplam-responsive and risdiplam-responsive exons from genes identified transcriptome-wide in the previously published datasets (Fig. 8b). From the combined set of pseudoexons (Supplementary data 1), SpliceAI identified primarily rare variants within the 50 base pairs (bp) adjacent to pseudoexon splice junctions (Supplementary data 2). Near

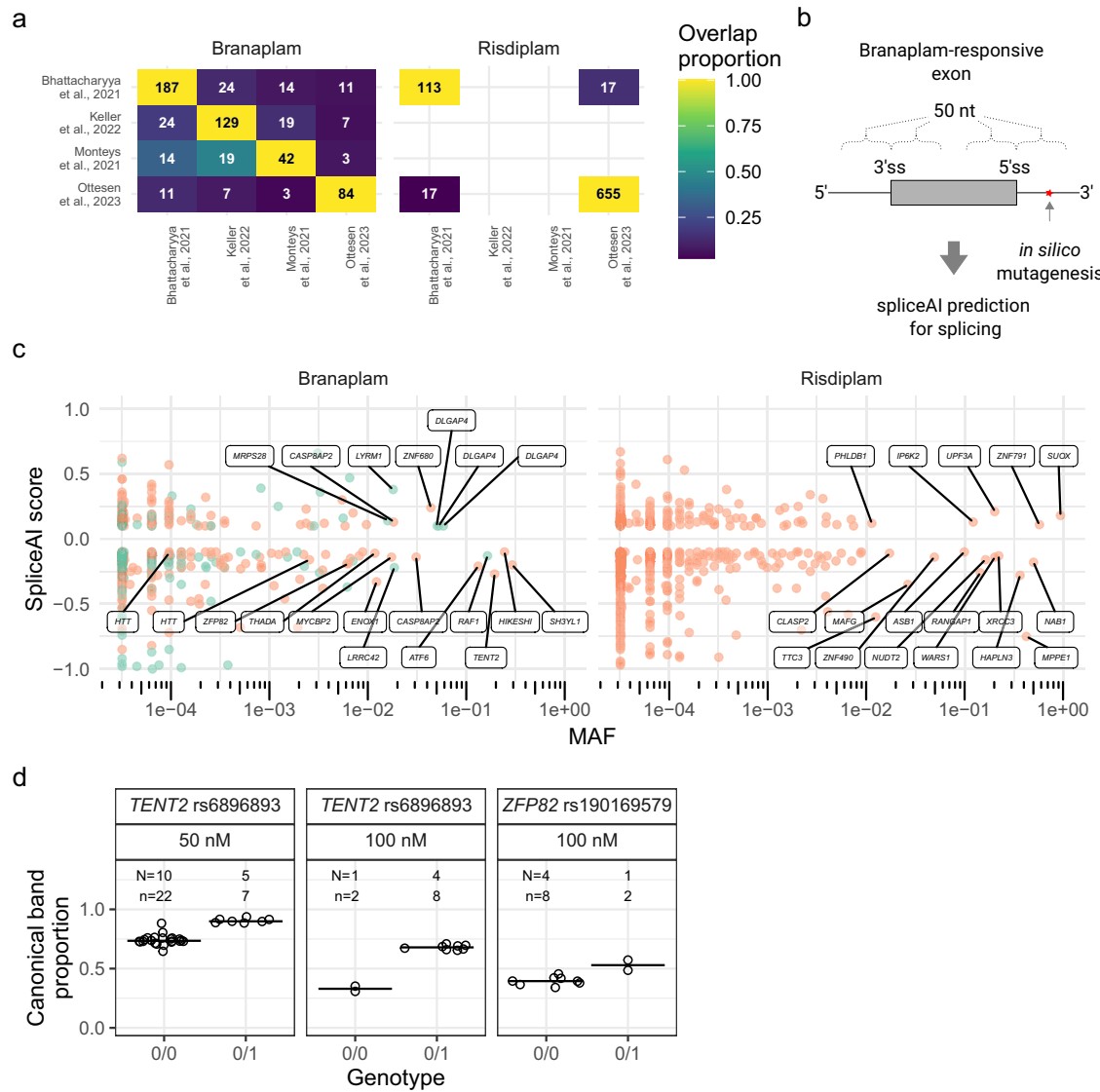

**Fig. 8 | SpliceAI identified variants predicted to affect splicing of branaplam-responsive exons genome-wide. a** The overlap of genes with drug-responsive exons in the different studies analyzed for branaplam and risdiplam treatment. For each comparison, the number of genes in common across each dataset (white number) was normalized on the total number of genes for each row (black number). The color indicates the overlap proportion (white number / black number). **b** SpliceAI predictions were made for variants within 50 nt of branaplam-responsive exon and pseudoexon splice junctions. **c** Variants near branaplam- (left) or risdiplam-responsive pseudoexons (orange) and exons (green) that yield

significant SpliceAI scores are plotted by allele frequency with gene names indicated for selected variants. *HTT* variants rs148430407 (MAF $2.6 \times 10^{-3}$) and rs772437678 (MAF $9.6 \times 10^{-5}$) are labelled, while rs145498084 did not produce a significant score. (**d**) SpliceAI-predicted variants affect splice modulation of *TENT2* and *ZFP82*. Proportion of canonically spliced product across tested LCLs for *TENT2* and *ZFP82*, grouped by absence (0/0) or heterozygous presence (0/1) of variant. N = Number of cell lines for variant, n = cultures analyzed. Source data are provided as a Source Data file.

the *HTT* pseudoexon, only rs772437678, which interferes with the branaplam effect, and rs148430407, which we were unable to test, yield significant negative SpliceAI scores, consistent with a reduction in pseudoexon inclusion. For *PMS1*, no surrounding polymorphic variants are predicted to affect splicing, with only very rare variants within 50 bp either side of the pseudoexon. At MAF > 1 %, single variants in other genes show significant SpliceAI scores. Across branaplam and risdiplam, seven variants are predicted to enhance the incorporation of a pseudoexon (positive SpliceAI score), sensitizing them to the splice modulation and making that gene a more sensitive target. Conversely, 20 variants are predicted to interfere with pseudoexon inclusion (negative SpliceAI score), which could make them refractory to drug-induced splicing and reduce the off-target impact. SpliceAI also predicted frequent effects from seven common variants (MAF >

1%; five positive and two negative SpliceAI scores) in five genes, among those reported to display drug-responsive alternative splicing of annotated exons. We validated the SpliceAI results by testing two variants predicted to have a negative effect on pseudoexon splicing probability in *TENT2*, (rs6896893, spliceAI score −0.27, MAF 19%) and *ZFP82* (rs190169579, spliceAI score −0.19, MAF 0.63%), respectively. First, we confirmed that branaplam treatment resulted in pseudoexon inclusion for both genes (Supplementary Fig. 12). When treated with 50 nM branaplam, LCLs heterozygous for the *TENT2* SNV showed less pseudoexon inclusion (i.e., a higher proportion of canonical transcript) than those homozygous for the major allele (0.91, 95% CI: 0.88 to 0.93 versus 0.74, 95% CI: 0.73 to 0.76; $p < 0.0001$) (Fig. 8d). Treatment with 100 nM branaplam further accentuated this effect (0.68, 95% CI: 0.66 0.69, versus 0.33, 95% CI: 0.30 to 0.36; $p = 0.0002$)

(Fig. 8d). Similarly, LCLs heterozygous for the *ZFP82* SNV treated with 100 nM branaplam showed a higher proportion of canonical *ZFP82* transcript, 0.53 (95% CI: 0.47 to 0.58) compared to 0.40 (95% CI: 0.37 to 0.42) ($p = 0.02$) in LCLs without the minor allele (Fig. 8d). Overall, our results indicate that polymorphic variation in the human population can have important implications for both on-target and off-target effects of small molecule splice modulators. SpliceAI analysis provides an initial step in the development of computational tools to adequately assess such drugs as they are developed for therapeutic use.

## Discussion

GWAS of HD age-at-onset and other clinical landmarks have identified several loci that encompass DNA maintenance genes as genetic modifiers that hasten or delay the disease[5,34]. Several of these are DNA repair genes also implicated as modifiers of CAG repeat instability in cell and animal models, contributing to the proposal that HD onset is driven by somatic expansion of the *HTT* CAG repeat[35]. For example, *FAN1* acts as a suppressor of repeat expansion[36,37] and damaging *FAN1* variants are associated with hastened onset[38]. *MLH1* encodes a mismatch repair protein in which a common missense variant is associated with delayed HD onset. Another mismatch repair gene, *MSH3*, displays both onset-hastening and onset-delaying haplotypes and is required for somatic CAG expansion[5,34,39,40]. In this context, a GWAS peak spread across ~200 kb of chromosome 2 spans four genes, *ANKAR*, *OSGELP1*, *ORMDL1*, and *PMS1*, and comprises two independent modifier effects, one HD hastening and one delaying[5,34]. The most significant SNPs in the modifier haplotypes are in *ANKAR* and *ORMDL1*, respectively, and no specific functional variant has been identified as responsible for either modifier effect. Instead, modification by these common haplotypes has been presumed to be due to sequence differences affecting *PMS1*, given its relationship with those DNA repair genes where functional variation has been clearly demonstrated[23,24,41]. A genetic modifier effect of *PMS1* has received support, albeit only at nominal significance, from the increased burden of predicted damaging *PMS1* variants in HD individuals with extremely delayed HD onset compared to those with extremely early onset[38]. Our demonstration that *PMS1* inactivation reduces CAG repeat expansion in a human cell line model bolsters *PMS1* as the likely source of the chromosome 2 locus HD modifier effects. Defining the functional polymorphic variants and the mechanism by which they alter *PMS1* expression will probably require specific analysis of neurons susceptible to CAG expansion from HD individuals. However, our findings suggest in any event that *PMS1* is a potential therapeutic target whose functional reduction, either by splice modulators or by other strategies, would provide a fundamentally different alternative to therapeutic strategies based on reducing mutant huntingtin, i.e., reducing the rate of somatic CAG expansion to delay or prevent onset.

Given their association with tumor formation, HD therapeutic strategies aimed at mismatch repair genes known to modify HD are not without potential risk. *PMS2* and *MLH1* are key genes whose inactivation causes Lynch syndrome, involving colorectal, endometrial, and other cancers[42]. Biallelic germline mutations in these genes cause constitutional mismatch repair deficiency (CMMRD) syndrome with a predisposition to tumor formation in childhood[43]. A more attractive target is *MSH3*, which is not an established cause of Lynch syndrome, although biallelic *MSH3* germline mutations have recently been associated with attenuated colorectal adenomatous polyposis and a phenotype less severe than CMMRD, without pediatric tumors[44–46]. Like *Msh3* and in contrast to *Mlh1* and *Pms2*, the loss of *Pms1* does not cause tumors in mice, albeit aged to only 12 months[47]. PMS1 physically interacts with MLH1 to form the MutL-Beta complex, but its role in canonical mismatch repair is unclear[48]. While it is not associated with Lynch syndrome, at least one case of CRMMD attributed to biallelic *PMS1* germline mutation[49], somatic loss of the wild-type allele in an individual with a germline deletion of *PMS1* and multiple tumor types[50], and reduced PMS1 in oral squamous cell carcinoma without *PMS1* mutation all point to the need to better understand the potential consequences of life-long downregulation of *PMS1* which could also have effects beyond DNA repair[51].

Also uncertain is the degree to which PMS1 would need to be lowered for therapeutic benefit. Our finding that heterozygous inactivation of *PMS1* does not reduce the amount of PMS1 or the rate of CAG repeat expansion in our cell model suggests that, at least in this context, the protein's level is normally regulated beyond gene dosage, perhaps by the stoichiometry of interaction with a limiting binding partner[23,51,52]. This contrasts with MSH3, where mouse model experiments indicate that while homozygous inactivation of *Msh3* eliminates somatic CAG expansion, even a 50% lowering of MSH3 is sufficient to cause a measurable reduction[27,40]. The lack of such a gene dosage effect for PMS1 implies that a therapeutic benefit in HD might require considerably more than 50% lowering of gene expression. However, the human GWAS data revealed a measurable delaying effect on HD age-at-onset due to a common haplotype not involving damaging variants or a strong *PMS1* expression quantitative trait locus (eQTL) association. This argues that factors regulating PMS1 expression and its effect on CAG repeat expansion specific to the neurons susceptible to HD may determine its modification of HD onset.

Like the effect of *PMS1* inactivation on the CAG repeat in our cell model, loss of PMS1 also largely prevents expansion of the CGG repeat in a mouse embryonic stem cell model of the fragile X-related disorders[53], suggesting that PMS1 has broader relevance as a potential target for therapeutic downregulation across repeat disorders. Interestingly, in our study and the mouse CGG repeat model, a small degree of repeat expansion remained after knocking out *PMS1/Pms1*. This residual expansion could limit the suppression of CAG repeat expansion achievable by downregulating PMS1, depending on whether it is due to the expression of an alternate PMS1 isoform in these models or functional redundancy in the mismatch repair machinery.

While the presence of a modulable pseudoexon in *PMS1* provides a route to achieve its downregulation via small molecule splice modulators, this strategy has multiple levels of complexity that must be considered. Orally-available small molecule splice modulators do provide an attractive option for therapeutic development, especially for genetic diseases of the nervous system. Their potential has been demonstrated by the United States Food and Drug Administration's (FDA) approval of risdiplam for the treatment of spinal muscular atrophy (SMA), where it promotes the inclusion of exon 7 in *SMN2*, whose product then compensates for *SMN1*-inactivating mutations. Branaplam was also tested in SMA patients[54]. However, the suspension of the branaplam clinical trial in HD due to safety concerns over the development of peripheral neuropathy[16] emphasizes the potential for side-effects due to the broad action of such splice modulators evident in the wide range of transcriptional effects of branaplam and risdiplam in published studies[14,15,31,32]. As we have demonstrated, an added layer of complexity is the impact of genetic variation in influencing drug effects at both target and off-target loci. For *HTT*, we identified rare variants that affected pseudoexon inclusion whose impact would depend on the chromosome carrying them. On the non-expanded *HTT* chromosome, the outcome might be positive, allowing continued expression of wild-type huntingtin, whereas on the expanded CAG chromosome, continued expression of mutant but lower expression of wild-type would be more likely to have a deleterious outcome. Another concern with genetic variation is the potential for unexpected off-target effects. We identified many such potential variants, most of which are very rare, but across many individuals, the likelihood of a patient with such a variant receiving drug is non-trivial. Our approach was biased, relying on known branaplam- and risdiplam-responsive exons. However, identifying novel pseudoexons activated by genetic variation would be an important next step. Clearly, human genetic variation should be considered with any therapeutic approach

targeting specific genetic sequences, whether mediated by proteins or nucleic acids, such as CRISPR-Cas modification[55] or by small molecules, as described here. Encouragingly, we show that AI tools can be used to identify genetic variants and potential off-targets, allowing an approach of screening patients before they receive such interventions.

While all of the above factors must be considered carefully in developing a potential therapeutic, these small molecule splice modulators have huge delivery advantages with their oral availability and broad distribution, including into the cortex and striatum[15]. Indeed, inherent in their differential potency and off-target effects is the promise that chemical modifications and a better understanding of the mechanism of splice modulation can identify compounds that more specifically target individual genes and reduce potential side-effects. The drugs are proposed to drive alternative splicing by stabilizing non-canonical nGA 3′-exonic motifs at the 5′ss[14,15]. Our results of editing the *HTT* pseudoexon 5′ss support that mechanism, with a single A insertion between the exonic and intronic splice motifs preventing pseudoexon splicing. However, this editing prevented both pseudoexon inclusion (exon 50a) and the generation of the alternative product (exon 50b) that does not use this pseudoexon 5′ss. The exon 50b product was detectable in the RNAseq results of previous publications[14,56], but was not focused upon since it results in the same frame-shifting outcome. We speculate that this product can fit within the nGA 3′-exonic motif stabilization model through the order of intron splicing and intron retention, which can be driven by the relative strength of the splice sites[57]. When we weakened the intron 49 upstream splice site in a minigene, we observed a decreased exon 50b product ratio relative to the exon 50a product. Additionally, the strong effect of genetic variants near the *HTT* pseudoexon 3′ splice site suggests an important role for this 3′ss region in the drugs' efficacy. There may also be alternative explanations, with the drugs having an unexplained component to their mechanism. Indeed, a recent publication challenges how branaplam interacts with the U1 / 5′ss, proposing two interaction modes, one for the nGA 3′-exonic motif stabilization and a second interaction with the surrounding sequence[58]. It also suggests that cocktails of the splice modulators show synergy and can influence the target specificity[58]. Together with further chemical modification, this synergy increases the options for identifying splice-modulating therapeutics that specifically target *PMS1* for repeat expansion disorders and, ultimately, that target other genes in diseases where modulating alternative splicing could prove beneficial.

Downregulation of *PMS1* by splice modulator promotion of pseudoexon inclusion represents just one approach for reducing *HTT* CAG repeat expansion. Indeed, our finding that inactivation of *PMS1* by CRISPR-Cas editing represents another attractive option and opens the way to other nucleic acid and small molecule-based approaches. For HD and other CAG repeat disorders, the cell line system that we have developed, which shows relatively rapid CAG expansion in confluent cultures, will facilitate the discovery, testing and development of approaches to develop therapeutic interventions that act on the initial somatic DNA expansion phase of pathogenesis.

## Methods

### Lymphoblastoid cell lines and drug treatment
This work complies with all ethical regulations as approved by the Mass General Brigham (MGB) Institutional Review Board (IRB). Lymphoblastoid cell lines (LCLs)[59] from HD patients had been generated in previous studies and were approved for secondary use in this study by the MGB IRB. LCLs were grown in suspension in RPMI 1640 medium (MilliporeSigma, 51536 C), with 15% fetal bovine serum (MilliporeSigma, F0926). For branaplam (Synonyms: LMI070, NVS-SM1) (MedChemExpress, HY-19620,) or risdiplam (Synonyms: RG7916; RO7034067) (MedChemExpress, HY-109101) treatments, a 1 mM stock solution prepared in DMSO was diluted in media to the concentrations indicated for 24 h. Each experiment had the same cell line treated as a

control, which was used to correct for run-run variation for the gel-based PCR quantification. LCLs of desired genotype were selected from participants in a published genome-wide association study[5]. The cell lines are available for non-commercial purposes upon request with a standard MGB material transfer agreement.

### RNA isolation, cDNA synthesis, PCR, and densitometry
RNA was isolated using TRIZOL reagent (Invitrogen, 15596026) following the manufacturer's protocol. Any contaminating genomic DNA was removed using ezDNase (Invitrogen, 11766051) following the manufacturer's protocol. The cDNA was synthesized using the Superscript IV kit (Invitrogen, 18091050) with poly(A) oligo(dT) with an incubation at 50 °C and 80 °C for 10 min each, followed by an incubation with RNase H at 37 °C for 20 min.

The relative pseudoexon inclusion was quantified by PCR from exons flanking the pseudoexon (Supplementary table 1). We used GoTaq G2 Hot Start PCR kit (Promega, M7423) with the following conditions: initial denaturation 94 °C (2 min), 40 cycles of 94 °C (30 s), 60 °C (30 s), 72 °C (45 s), final extension 72 °C (5 min). Amplicons were loaded onto a 2% agarose gel with EZvision (VWR, 97064-190), and the band intensity was quantified by densitometry using ImageJ[60].

### Minigene cloning, mutagenesis, and transfection
A minigene construct was prepared by isolating the entire *HTT* exon 49–50 region of interest (Supplementary Table 1) from HEK293T genomic DNA. We used the Q5® High-Fidelity PCR Kit (New England Biolabs, E0555S) with the following conditions: initial denaturation 98 °C (3 min), 35 cycles of 98 °C (10 s), 64 °C (30 s), 72 °C (60 s), final extension 72 °C (2 min). This PCR fragment was TOPO cloned into pcDNA™3.1/V5-His backbone (Invitrogen, V81020). We used in vivo assembly cloning[61,62] for site-directed mutagenesis to modify the nucleotide 1 bp upstream of the exon 49 splice junction to each of the alternative nucleotides (Supplementary Table 1). The PCR for cloning was with UltraRun® LongRange PCR Kit (QIAGEN, 206442) with the following conditions: initial denaturation 93 °C (3 min), 18 cycles of 93 °C (30 s), 60 °C (15 s), 68 °C (3 min 35 s), final extension 72 °C (10 min). The amplicons were treated with DpnI restriction enzyme to remove the plasmid template and transformed into XL10 gold competent cells prepared by Mix and Go! transformation kit (Zymo Research, T3001). The sequence of the isolated plasmids was confirmed using nanopore sequencing (Plasmidsaurus, SNPsaurus LLC). Confirmed plasmids were transfected into HEK293T cells with lipofectamine 3000 (Invitrogen, L3000001) following the manufacturer's protocol.

### ddPCR gene expression quantification
Absolute expression quantification was carried out with the QX200 Droplet Digital PCR (ddPCR, Bio-Rad). We used the primer mix for probes (no dUTPs) (Bio-Rad, 1863023) and AutoDG Instrument (Bio-Rad, 1864101) for automated droplet generation following the manufacturer's instructions. All primers and probes are listed in Supplementary Table 1.

### RPE1-AAVS1-CAG115 model generation
The RPE1-AAVS1-CAG115 model was generated by targeted knock-in of a *HTT* exon1 fragment into the AAVS1 safe harbor locus. We isolated the entire exon 1 of *HTT* with 115 CAG repeats from an HD patient with UltraRun® LongRange PCR Kit (QIAGEN, 206442) with supplementation of 10% DMSO under the following conditions: initial denaturation 93 °C (3 min), 35 cycles of 93 °C (30 s), 61 °C (15 s), 68 °C (60 s), with a final extension of 72 °C (10 min). The primers (Supplementary Table 1) had flanking SalI sites which were used to insert the *HTT* fragment as a GFP fusion-protein (Supplementary Fig. 13) in an all-in-one tetracycline-inducible expression cassette with AAVS1 homology arms (AAVS1-TRE3G-EGFP was a gift from Su-Chun Zhang (Addgene

plasmid # 52343; http://n2t.net/addgene:52343; RRID:Addgene_52343). This plasmid contains a promoterless puromycin resistance gene with a 3' splice site that generates puromycin resistance when correctly inserted into intron 1 of *PPP1R12C* (also known as AAVS1)[63]. hTERT RPE-1 (CRL-4000 – ATCC, female origin) were transfected with lipofectamine 3000 (Invitrogen, L3000001) following the manufacturer's protocol with AAVS1 targeting vector and predesigned transcription activator-like effector nucleases (hAAVS1 TALEN Left and Right were gifts from Su-Chun Zhang, Addgene plasmid # 52341 & 52342; http://n2t.net/addgene:52341; http://n2t.net/addgene:52342; RRID:Addgene_52341; RRID:Addgene_52342). Since hTERT RPE-1 already has expression of puromycin resistance gene, we selected with a high 20 µg/mL dosage of puromycin for 1 week. Clones were isolated by limited dilution and screened for presence of transgene insertion by PCR of the 5' homology arm over the puromycin resistance gene (Supplementary Table 1). We used GoTaq G2 Hot Start PCR kit (Promega, M7423) with the following conditions: initial denaturation 94 °C (2 min), 35 cycles of 94 °C (30 s), 60 °C (30 s), 72 °C (60 s), final extension 72 °C (5 min). The RPE1-AAVS1-CAG115 cell line is available for non-commercial purposes upon request with a standard MGB material transfer agreement.

### Edu (5-ethynyl-2'-deoxyuridine) cell cycle analysis
RPE1-AAVS1-CAG115 cells were seeded onto coverslips at densities decreasing by half, allowing the cultures to reach confluency on consecutive days. The cultures were treated with Edu for one hour and processed using Click-iT™ EdU Cell Proliferation Kit (Alexa Fluor™ 488 dye, Invitrogen, C10337) following the manufacturer's instructions. The coverslips were mounted and 160 images per group were captured using the Celldiscoverer 7 (Zeiss) for automated epifluorescent image acquisition. Images were analyzed using CellProfiler[64] and StarDist[65] to identify the nuclei in the dense cultures, and CellProfiler Analyst[66] for machine learning-based categorization of Edu-positive cells.

### Flow cytometry
Cells were dissociated, filtered through a 40 µm cell strainer, and sorted/analyzed for GFP signal using BigFoot Cell Sorter (Thermo-Fisher). The data were analyzed in R using the flowCore[67], flowTime[68], and flowAI[69] packages.

### PMS1 western blot
RPE1-AAVS1-CAG115 cells grown to confluency in 6-well plates were rinsed once with ice-cold PBS and lysed in situ with 100 µl per well of ice-cold RIPA buffer (Boston BioProducts, BP-115) containing Pierce™ protease and phosphatase inhibitor cocktail (Thermo Scientific, A32959). Lysates were incubated on a rotary mixer for 30 min at 4 °C followed by centrifugation at 16000 g for 10 min (4 °C) and collection of the supernatant. Protein concentration was measured in the extracts with Pierce™ BCA assay kit (Thermo Scientific, 23225). Protein extracts (20 µg/lane) were resolved on NuPage™ 10% Bis-Tris (Invitrogen, NP0303) or 3–8% Tris-acetate mini gel (Invitrogen, EA03755) and transferred to 0.45 µm nitrocellulose membrane (Thermo Scientific, 88018). The membrane was blocked for 1 h at RT in 5% non-fat dried milk in TBST (Tris-buffered saline with 0.1% Tween 20). Primary antibodies: PMS1 (mouse monoclonal 68413-1-Ig, Proteintech, 1:4000), β-actin (rabbit polyclonal 4967 S, Cell Signaling Technology, 1:1000), in blocking solution, were applied overnight at 4 °C. Secondary antibodies were horseradish peroxidase-linked anti-mouse and anti-rabbit IgG (NA931 and NA934, respectively; GE Healthcare), both 1:4000 in blocking solution, incubated for 2 h at RT. Western blots were developed with Pierce™ ECL kit (Thermo Scientific, 32106).

### Cytotoxicity analysis
Acute cytotoxicity was quantified in RPE1-AAVS1-CAG115 cells with Incucyte SX5 (Sartorius) high throughput image analysis. Cells were seeded at 5000 cells per well and imaged every 2 h for three days. We also treated with Incucyte® Cytotox Red Dye (Sartorius, 4632) following manufacturer's instructions. The cell confluency and count of cytotox-stained nuclei was quantified using the Incucyte software.

For cytotoxicity in long-term culture, we grew the cells to confluency and treated with selected drug concentrations for two weeks. We treated with Incucyte Cytotox Red Dye and analyzed the cells after 20 h. We used the python scikit-image package[70] to count the number of dead cells as well as quantify the background autofluorescence. For counting dead cells, we set a threshold and segmented stained nuclei with a minimum object size of 5 pixels to exclude artifacts. For the autofluorescence analysis, calculated the mean pixel intensity above the background but below the threshold used to identify the stained nuclei.

### Repeat instability analysis
We carried out CAG repeat instability experiments with a high-throughput plate-based pipeline from growing the cells all the way through to capillary electrophoresis. The RPE1-AAVS1-CAG115 were seeded into 96-well plates and grown to confluency to trigger contact inhibition, which enables analysis of repeat expansion in the absence of cell division. The cells were fed every 2–3 days for a total of 4–6 weeks, with genomic DNA isolated using the Quick-DNA 96 Kit (Zymo Research, D3011).

Repeat tracts were quantified by PCR amplification followed by capillary electrophoresis. We used the Taq PCR Core Kit with Q solution (Qiagen, 201225) with 5 µL of the isolated genomic DNA following PCR conditions: initial denaturation 95 °C (5 min), 30 cycles of 95 °C (30 s), 65 °C (30 s), 72 °C (1 min 30 s), final extension 72 °C (10 min). We optimized the PCR with the nested design to only amplify the transgenic exon 1 fragment, which we used for the instability experiments following pseudoexon editing. This PCR had an outer amplicon (Supplementary Table 1) for 12 cycles under the same conditions above, followed by the standard fragment analysis assay for the inner amplicon with an additional 22 cycles. Amplicons were analyzed using a 3730XL DNA Analyzer (36 cm array, POP-7 Polymer, standard fragment analysis conditions) with 0.8 ul PCR product is loaded in 9.4 ul Hi-Di Formamide (Applied Biosystems), with 0.1 ul GeneScan 500 LIZ (Applied Biosystems). The fragments were identified and converted to bp sizes using GeneMapper 5.0 (Applied Biosystems). Repeat length for each fragment within a sample was calculated from linear models fit using samples with known repeat lengths for each run.

The GeneMapper output was used to calculate repeat instability metrics using a custom pipeline in R, which we have organized as a package [https://doi.org/10.5281/zenodo.10825847] (https://github.com/zachariahmclean/instability). We calculated a repeat instability metric, average repeat gain, describing the average number of repeat units a population of repeat fragments changes from a defined starting point, similar to what was described previously[71]. We first defined a window of 40 repeat units to either side of the identified modal repeat for each sample, with a fragment height threshold of 5% of the modal repeat height. The weighted repeat length was then calculated for each sample by finding the weighted arithmetic mean of the CAG repeat length using the peak height as the weighting. The average repeat gain was the difference between the weighted repeat length for a timepoint and the starting timepoint. When there were multiple timepoints, average repeat gain per week was calculated by fitting a linear modal with a fixed intercept through the average repeat gain at time 0, then finding the slope. With just one timepoint, the average repeat gain was divided by the number of weeks.

## Genome editing

Various CRISPR-Cas9 approaches were used for genome editing experiments in RPE1-AAVS1-CAG115 cells. We used CRISPick[72,73] to select gRNAs (Supplementary Table 2). For the HD modifiers, we cloned oligos encoding the spacers of the gRNAs into pSpCas9(BB)-2A-Blast, which was a gift from Ken-Ichi Takemaru (Addgene plasmid # 118055; http://n2t.net/addgene:118055; RRID:Addgene_118055). The plasmids were transfected into RPE1-AAVS1-CAG115 using the 4D-Nucleofector X Unit (Lonza) and the P3 4D-Nucleofector™ X Solution (V4XP-3024) following the manufacturer's protocol and the EA-104 Nucleofector program. The cells were treated with 25 µg/mL Blasticidin for 4 days, followed by an additional 10 µg/mL for 7 days selection. To amplify *FAN1*, *MSH3*, and *PMS1* (Supplementary Table 2) target sites, we used the Q5® High-Fidelity PCR Kit (New England Biolabs, E0555S) with the following conditions: initial denaturation 98 °C (3 min), 35 cycles of 98 °C (10 s), 60 °C (30 s), 72 °C (60 s), final extension 72 °C (2 min). We pooled amplicons from the four different genes together and sequenced with llumina MiSeq via the MGH Center for Computational and Integrative Biology DNA core. CRISPResso pooled[74] was used to demultiplex the reads and quantify editing outcomes.

The polyclonal cell populations were found to be edited with 83%, 33%, and 57% indels for *FAN1*, *MSH3*, and *PMS1*, respectively (Supplementary Fig. 4a). The most common edits in each population were single bp insertions for *MSH3* (25% of reads) and *PMS1* (43% reads), but for *FAN1*, the most common edit was a 99 bp deletion (16% of reads). These edits resulted in frameshift in 38% *FAN1*, 32% *MSH3*, and 56% *PMS1* of reads. The *FAN1* population had a large number of deletions, with 46% of reads having a > 20 bp deletion, compared to an average 0.7% for the other targets. We analyzed the effect of these perturbations in a 6-week repeat instability experiment. The modal repeat lengths for the initial populations were very similar, with 127 repeats for non-targeting control and *FAN1*, 126 for *MLH3* and 125 for *PMS1*.

To analyze the *MSH3* and *PMS1* clonal strains from these edited pooled populations, we genotyped the clones with a barcode multiplexing strategy. Up to eight samples were uniquely barcoded with a unique identifier sequence on the forward primer, with the amplicons pooled, sequenced as described above, demultiplexed in silico, and each clone's read analyzed with CRISPResso. Clones were called homozygous when the top editing outcome accounted for more than 85% of the two most frequent aggregated editing outcomes, otherwise, they were called heterozygous.

To precisely target the pseudoexon location, we manually selected gRNA sequences with predicted cut sites within 3 bp of the splice site. We cloned oligonucleotides encoding the gRNA spacers into BPK1520 (Addgene plasmid # 65777) to generate gRNA expression plasmids. These plasmids were co-transfected with wild-type SpCas9 (RTW3027, Addgene plasmid # 139987) or the SpG variant capable of targeting sites encoding NGA PAMs (RTW4177, Addgene plasmid # 139988) (Supplementary Table 2). The plasmids were transfected with nucleofection as described above and GFP positive cells were FACS sorted with FACSAria™ III Cell Sorter (BD Life Sciences) 48 h after transfection. The editing was quantified by Sanger sequencing trace decomposition[75] and confirmed by sanger sequencing ion the isolated clonal strains by Sanger sequencing. For *PMS1* deletion of pseudoexon, two gRNAs flanking the 5' pseudoexon splice site were transfected as described above with the pSpCas9(BB)-2A-Blast vector. Clonal cell strains were screened for deletion by PCR with primers flanking the *PMS1* pseudoexon location (Supplementary Table 1).

## Predicting the effect of variants on pseudoexon splicing

To predict the effect of genetic variation on all known genes with pseudoexons, we used pseudoexons identified from RNAseq in four publications[14,15,31,32] and a previously described approach[76]. For Ottesen et al., intron retention events were excluded to be more comparable to the other studies. Briefly, sequences were taken 50 bp to either side of each of the pseudoexon splice sites, with in silico saturation mutagenesis to modify each position to the other three alternative nucleotides, followed by using spliceAI[33] to predict the effect of each variant on pseudoexon splicing based on the flanking exons of the gene. To determine if the exon is annotated, we compared the exon positions to the Ensembl GRCh37 version 75 annotations with a 1 bp tolerance.

## Statistics

The data were analyzed with R[77] and the tidyverse suite of packages[78], and marginaleffects[79]. P-values are the result of two-tailed t-tests. All data, graphs and statistics are available with executable R code (https://github.com/zachariahmclean/2023_splice_modulators).

## Reporting summary

Further information on research design is available in the Nature Portfolio Reporting Summary linked to this article.

## Data availability

The data supporting the findings of this study are available from the corresponding authors upon request. Source data are provided with this paper. All data, graphs and statistics are also available with executable R code (https://github.com/zachariahmclean/2023_splice_modulators). The SpliceAI data used for predictions (Supplementary data 1) and results (Supplementary data 2) generated in this study are provided as supplementary data files. The SpliceAI data used for predictions (Supplementary data 1) in this study are available in the following publications: Keller et al.[15] [https://doi.org/10.1038/s41467-022-28653-6], Bhattacharyya et al.[14] [https://doi.org/10.1038/s41467-021-27157-z], Monteys et al.[31] [https://doi.org/10.1038/s41586-021-03770-2], and Ottesen et al.[32] [https://doi.org/10.1093/nar/gkad259]. Source data are provided with this paper.

## Code availability

The code we used for analyzing CAG repeat instability from fragment analysis traces is available as an R package (https://github.com/zachariahmclean/instability).

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

## Acknowledgements

Supported by Hereditary Disease Foundation Fellowships (Z.L.M. and J.C.L.R.), NIH grants NS091161 (J.F.G.), NS126420 (R.M.P.), NS049206 (V.C.W.), NS105709 (J-M.L.), NS119471 (J-M.L.) and DP2-CA281401 (B.P.K.), an MGH ECOR Howard M. Goodman Fellowship (B.P.K.), the CHDI Foundation (J.F.G., M.E.M.), and the Huntington's Disease Society of America Human Biology Project (R.M.P.). These HD studies would not be possible without the vital contribution of the research participants and their families.

## Author contributions

Z.L.M.: Conceptualization, Methodology, Software, Formal analysis, Investigation, Data Curation, Writing - Original Draft, Visualization, Project administration. D.G.: Software and Formal analysis prediction of variants on splicing. K.C.: Software for repeat instability, cytotoxicity image analysis, genotyping, and phasing. J.C.L.R.: Methodology modifier CRISPR-Cas9 development. S.S.: Methodology and Software for modifier CRISPR-Cas9 development sample genotyping. I.N.F.: Investigation Fig. 3. Z.E.N.V.M.: Investigation Fig. 2. M.K.: PMS1 western blotting. M.R.: Software cytotoxicity analysis. E.M.: Resources minigene cloning vector, Critical Reading. J.R.: Experimentation cell culture, Resources LCLs. T.G.: Experimentation CAG sizing, sequencing. D.L.: Resources human subjects. B.P.K.: Methodology and Resources CRISPR-Cas9 pseudoexon editing Fig. 7. J.M.L.: Resources identification of LCLs. M.E.M.: Resources identification of LCLs, Supervision, Critical Reading. V.C.W.: Conceptualization, Resources. R.M.P.: Conceptualization, Resources, Methodology modifier CRISPR-Cas9 development and sample genotyping. J.F.G.: Conceptualization, Resources, Writing - Original Draft, Supervision, Project administration, Funding acquisition.

## Competing interests

J.F.G. and V.C.W. were founding scientific advisory board members with a financial interest in Triplet Therapeutics Inc. Their financial interests were reviewed and are managed by Massachusetts General Hospital (MGH) and Mass General Brigham (MGB) in accordance with their conflict of interest policies. J.F.G. consults for Transine Therapeutics, Inc. (dba Harness Therapeutics) and has previously provided paid consulting services to Wave Therapeutics USA Inc., Biogen Inc. and Pfizer Inc. V.C.W. is a scientific advisory board member of LoQus23 Therapeutics Ltd. and has provided paid consulting services to Acadia Pharmaceuticals Inc., Alnylam Inc., Biogen Inc., Passage Bio and Rgenta Therapeutics. R.M.P. and V.C.W. have received research support from Pfizer Inc. B.P.K. is a consultant for EcoR1 capital and Novartis Venture Fund, and is on the scientific advisory board of Acrigen Biosciences, Life Edit Therapeutics and Prime Medicine. B.P.K. has a financial interest in Prime Medicine, Inc., a company developing therapeutic CRISPR-Cas technologies for gene editing. B.P.K.'s interests were reviewed and are managed by MGH and MGB in accordance with their conflict-of-interest policies. J-M.L. consults for Life Edit Therapeutics and serves on the scientific advisory board of GenEdit Inc. E.M. is inventor on an International Patent Application Number PCT/US2021/012103, assigned to Massachusetts General Hospital and PTC Therapeutics entitled "RNA Splicing Modulation" related to use of BPN-15477 in modulating splicing. E.M. is a scientific advisory board member of ReviR Therapeutics, Inc and has received research support from PTC Therapeutics, Inc. The other authors declare no competing interests.
