## [Peer Review File · Nature Communications]

Splice modulators target PMS1 to reduce somatic expansion of the Huntington's disease-associated CAG repeatREVIEWER COMMENTS

Reviewer #1 (Remarks to the Author):

Review

This manuscript presents an interesting finding about the role of splice modulation in modulation of both HTT mRNA levels as well as somatic expansion of mHTT CAG repeat length. The authors present a compelling case for the role of the splice modulator drugs risdiplam and branaplam in the regulation of inclusion of a pseudoexon in PMS1 in addition to their reported role in regulation of inclusion of a translational inhibiting pseudoexon in HTT as well as describing two variants of the drug-induced alternative splicing of HTT. Their investigation of variant alleles in proximity to the HTT pseudoexon and the impact these variants have on activity of the splice modulators further supports the observations of several groups regarding the breadth of splicing biology impacted by these molecules. As relates to PMS1 alternative splicing, the authors show – consistent with the cited reports of PMS1 variants regulating somatic CAG repeat expansion – that drug-induced alternative splicing of PMS1 can regulate the rate of CAG expansion in an in vitro model, suggesting that treatment of HD patients with splice modulator drugs may have salutary effects via both inhibition of HTT mRNA / protein expression and the degree of somatic mHTT CAG repeat expansion. Overall, the work is quite interesting and represents interesting developments around both the MoA of splice modulator drugs and validation of the MoA of modifiers of mHTT pathology. It would be interesting (but beyond the scope of this manuscript) to see this work expanded into models of mutant HTT either in vitro or in vivo to further validate the impact of PMS1 and alternative splicing in regulation of the CAG repeat expansion and disease pathology.

One area of the manuscript that seems misplaced, or not well connected to the overall 'story' is the section featuring a discussion of the genome-wide SpliceAI predictions and alternative splicing by branaplam. This is not well integrated with the role of splice modulator drugs on HTT and PMS1 alternative splicing and their impact on the function and potential pathogenesis of HD. In addition, the work – while advancing from the variant allele data from the HTT transcript to predict additional alternative splicing events induced by branaplam – seems to provide little novel information beyond the several studies reported on transcriptome-wide alternative splicing induced by these drugs. I would suggest either better integrating this section into the overall section regarding HTT variant alleles (and contrasting with the noted lack of variant alleles around PMS1), or reserving this data for an independent manuscript that explores the role of these variant alleles and potential on-mechanism / off-target consequences of treatment with splice modulator drugs.

Overall, I would recommend the publication – following addressing the above noted major change and some minor notes below.

Minor notes / corrections:

Line 46 – sentence should start 'These features' (assuming this is referring to 'some DNA repair genes')

Line 72 – suggest including 'pharmacodynamic' in the sentence 'We found that the (pharmacodynamic) effectiveness of...

Line 273 – should include 'to' in the sentence ...figure 8b), predicted (to) result in ...

Line 341 – FDA is the 'Food & Drug Administration'

Figure 2A – suggest including a box – line figure showing HTT exons 49 and 50 (and the pseudoexon) to better highlight the variants and where they are relative to the exons

Reviewer #2 (Remarks to the Author):

McLean et al address a topic that is of great interest for biology, potential and ongoing therapeutics,

and is timely. They report some of the first data of an effect of PMS1 on CAG instability, as well as a complex effect of two splice modulators, branaplam and risdiplam, upon CAG instability. This study broadens the awareness of the many genes that can be targeted by these splice modulators, and the downstream effects that they may have. They identify variants of HTT that affect the efficacy of the splice modulators. They present a predictor to identify such splice site variants – properly developed this could be of great utility to all splice-modulation approaches. They begin to extend the action of branaplam and risdiplam to PMS1 splice modulation. One of the more exciting advances of this study is the beginnings of the establishment of a new model for CAG repeat instability. The cell shows an effect of transcription across the repeat, albeit the reverse of what is expected. The cell shows the expected suppressed and enhanced CAG expansions by knock-outs of MSH3 and FAN1, respectively. This study provides the first data supporting a role of PMS1 in enhancing CAG expansions. While the effect of PMS1 is milder than MSH3, the finding has implications for the biology of CAG instability and may have implications for therapeutics. Interestingly, the authors tested whether branaplam and risdiplam treatment affected CAG instability. This was based upon the suggestive claims of a bioRxiv submission of >3 years ago (their ref 18), of an unsubstantiated finding that lowering of HTT protein affects CAG instability, the authors tested whether branaplam and risdiplam treatment might affect somatic CAG instability. They reveal that branaplam treatment can slow CAG expansion rates in the cell model. They rule-out an effect of HTT splice modification in splice-modulator altered CAG expansions, and provide some support for a role of PMS1 splice modification in reducing CAG expansions.

While there are some potentially exciting novel findings in this study, there are numerous serious concerns outlined below:

Major concerns:

The cell line is an exciting advance, showing biased expansions over short culture times and expected increased and reduced expansions by knockouts of FAN1 and MSH3, respectively. Careful analysis and awareness of the model system will support the utility and relevance of the RPE1-AAVS1-CAG115 cell model as “a useful model for functional genomic investigations of CAG repeat instability”, as claimed.

1. The new model of CAG instability requires detailed characterization for its reflection of what is occurring in HD patient cells/tissues, or tissues of HD mouse models. For example, i) what is the endogenous AAVS1-driven direction of transcription across the CAG/CTG repeat? Is transcription direction the same as in the endogenous HTT gene? A schematic figure with promoters, repeat strands, and primers, would be helpful; ii) What is the inducible direction of transcription? Again, a schematic would be helpful; iii) Does the absence of transcriptional induction (doxycycline) result in an absence of transgene transcription, or only a reduced level? iv) Is instability affected by the absence of cell proliferation? v) The G0/1 arrest of cell proliferation under the test conditions used must be demonstrated; vi) If claims are to be made or inferred, as it is, that instability, in such small, 96-well cultures, arise during G0/1, the amount of cell proliferation following seeding to G0/1 arrest should be determined. vii) It appears that there is proliferation, since the description of the results in Fig. 5 state that the cells were maintained at confluency, but increasing doses of branaplam decreased cell proliferation. This can be of concern if instability in the cell model occurred during proliferation.

1a. While not necessary for cell model establishment, it would be interesting to know viii) Are the effects of each tested modifier (MSH3, FAN1, PMS1) dependent upon transcriptional alteration across the expanded repeat? ix) Do the modifiers require absence of cell proliferation to alter instability?

2. While the AAVS1 locus has been suggested as a safe harbour since it is permissive for stable transgene expression, recent evidence reveals that considerable inter-clone variability of AAVS1 knock-ins does occur, as does transgene silencing, demanding careful attention to transgene expression (Bhagwan JR et al, 2019, F1000Res, 8:1911). Since transcription across the expanded repeat is well-established as a major driver of CAG instability, at a minimum it is crucial that levels of the transgene transcript be quantified under the various test conditions used.

3. Oddly, under non-induced/no-transcription across the repeat of the transgene, showed greater CAG expansions compared to induced conditions (Fig 4A). This is in opposition to many published findings where transcription across the expanded repeat enhances repeat instability (this is true in many model systems, many organisms, and HD mice, at the level of pol II elongation) and flies in the face of the ubiquitously expressed WT and mutant HTT in patients brains. This is very strange and should be addressed relative to published findings and relevance. This concern strengthens the need to characterize the models as outlined above (points 1 & 2). The authors may wish to revisit the Goula et al studies and interpretations (Goula et al, 2012, PLoSGenet 8:e1003051; Goula et al, 2013, Transcription, 4: 172-6).

3a. For the induced versus non-induced conditions, raw GeneMapper profiles should be provided (multiple clonal lines, not just pooled cells), rather than the batch processed data of Fig 4A.

4. The data that rule-out a role of drug-induced CAG instability through HTT splicing (and lowered HTT protein) is strong. Impressive, crafty, and clean experimentation. Nice! The authors should comment on this finding relative to the suggestion that lowering HTT protein modulates somatic CAG expansions (Coffey et al, their citation #18). It is possible that the effect observed by Coffey et al is likely due to ASO-induced transcriptional arrest at the HTT gene, as has been observed (Lee & Mendell, 2020, MolCell, 77:1044-1054; Lai F et al., 2020, MolCell, 77:1032-1043; Nakamori M et al., 2011, MolTher, 19:2222-7) and shown to reduce somatic repeat instability (Nakamori M et al., 2011, MolTher, 19:2222-7). Similarly, recent evidence showing that the near-complete silencing of HTT with di-valent anti-HTT siRNAs had no measurable effect on somatic repeat expansions in HD mouse brains (O'Reilly et al, 2023, MolTherap).

5. In contrast, the data supports some ability to connect, in part, the risdiplam-induced reductions in CAG expansions to altered PMS1 splicing, LIG1, FAN1, MLH1, MLH3, or any of the many DNA repair or other genes altered by their treatment (Bhattacharyya et al 2021 NatComm; Ottesen et al 2023 NAR). Is not possible without further experiments to be anything but suggestively connected to PMS1 altered splicing.

5a. Because the published effects of branaplam and risdiplam on LIG1, FAN1, MLH1, MLH3, and many DNA repair genes, but not of PMS1 (Bhattacharyya et al 2021 NatComm; Ottesen et al 2023 NAR) it is critical that the authors acknowledge that the effects may vary between cell lines/types. For example those studies also did not find any effect of those drugs on the levels of PMS1, however, McLean and colleagues do find PMS1 differences in their branaplam-treated LCLs. Thus other genes not affected in one cell line, may be affected in another. Moreover, McLean et al did not assess the effect of branaplam or risdiplam on PMS1 levels in the RPE1-AAVS1-CAG115 cell line they used for CAG instability. At a minimum, the authors must assess these.

5b. The effect of branaploam and risdiplam upon PMS1 splicing in LCLs is convincing. This must be done in the RPE cells under test conditions, and must be followed-up with evidence for transcript and PMS1 protein reduction. A western blot is a must.

5c. Line 331-6: "Overall, the results of targeting PMS1 via the drug inducible pseudoexon explained the reduction in rate of CAG repeat expansion caused by branaplam but only partially explained the observed effect with risdiplam. The partial effect with the latter along with the increases in expansion with the lower branaplam doses, suggest that the drugs may also have effects on splicing in other genes that influence CAG repeat instability." These authors cannot conclude that the branaplam-induced alterations in CAG expansions are explained by targeted PMS1 splicing. Both branaplam and risdiplam targeting of PMS1 splicing can only partially explain the altered CAG expansion rates induced by these drugs. The data do suggest that the effect of branaplam more directly involves PMS1 than does risdiplam.

6. The HD clinical trial of branaplam was halted due to concerns of safety and off-target splice modulation. Recent evidence for branaplam and risdiplam reveals that “both compounds triggered massive perturbations of splicing events, inducing off-target exon inclusion, exon skipping, intron retention, intron removal and alternative splice site usage” (Bhattacharyya et al 2021 NatComm; Ottesen et al 2023 NAR). Notably risdiplam was reported to downregulate a massive number of genes, including those that are known to affect CAG instability (MLH1, MLH3, FAN1, LIG1, and FANCI) cell cycle regulation, DNA replication, base excision repair, homologous recombination, and more (Bhattacharyya et al 2021 NatComm; Ottesen et al 2023 NAR). An analysis of the effects of branaplam and risdiplam treatments upon protein levels of MLH1, MLH3, PMS2, FAN1, LIG1, and PMS1 would minimize concerns of off-target effects over PMS1, in their role in splice-modulator induced CAG instability.

6a. The concerns of the off-target effects of the modulators demands an ability to predict their targets. Excitingly, McLean et al have developed this prediction using SpliceAI. The data in Figure 3 are predictions of branaplam, but not risdiplam, target sites based upon some, but not all of the published transcriptome-wide data. Specifically, the authors have selectively used the branaplam data from Monteyes et al. 2021; Bhattacharyya et al., 2021; Keller et al.,) and applied these to SpliceAI. They need to include the most current dataset of Ottesen 2023 (NAR), a study that they cite (their ref #33) for LIG1, which, amongst all the other hits of that study, was not included in the SpliceAI learning set. The branaplam data of the Ottesen study should be included.

6b. The authors make claims for both branaplam and risdiplam, but only use the published branaplam transcriptome-wide data for predictions and have neglected to apply the parallel risdiplam data for SpliceAI predictions. This should be done.

7. Contractions are under-whelming. Regarding claims of contractions for MSH3^{-/-}, the strength of this claim should be supported by continuous contractions in multiple clonal lines over longer term cultures. Contractions are not evident in the GeneMapper profile of the clonal MSH3^{-/-} line (Fig. S4B). An average repeat loss of 0.037 repeat units/week derived from 11 clones 22 cultures, for 0 and 28 days, equates to an average loss of 1 repeat over 27 weeks, or a loss of 2 repeats over just more than a year. Seems that contraction events were likely evident in only a handful of clones/cultures, as is evident by the distributions (Fig. 4E). If true, this is very very low levels. What appears as contractions in pooled cells (Fig. 4C) is, as the authors suggest, likely due to a mixture of edited and non-edited cells. But could also be due to culture over-take by cells with shorter repeats. In vivo data in HD mice did not reveal repeat contractions in any tissue, but might have missed these in the short life of a mouse.

8. The limited size and rate of expansions in the PMS1^{-/-} lines is convincing, as they seem to be biased for each line (Fig. 4E) and are readily evident in the clonal line (Fig. S4B). Data of Fig. S4B should have dots aligning the upper and lower repeat profiles, as done in Fig. 4C. This would facilitate an appreciation of the limited expansions.

9. Line 356: the statement “Additionally, in contrast to Mlh1 and Pms2, the loss Pms1 does not cause tumors in mice 26.”, should be specific, “Additionally, in contrast to Mlh1 and Pms2, the loss Pms1 does not cause tumors in mice, aged to only 12-months 26.” Confidence of safety should be toned-down. Some mention of the published known effects of human cancers and PMS1 or MSH3 heterozygosity, LOH, and homozygous mutations (somatic and germline), CMMRD, must be covered. For example, see (Hamad & Ibrahim 2022, HeredCancerClinPract, 20:16; Alghamdi et al, 2023, JEndocrSoc, 7:bvad035; Wang et al, 2013, Gene, 524:28-34). Clearly, while germline and somatic mutations in PMS1 may be rare in humans, they are not without serious life-impacting effects. Targeted reduction or inhibition of PMS1 for life-long administration, as would be needed for HD, could well have deleterious effects.

10. Line 361-: The authors have selectively mentioned only that damaging PMS1 variants in exome

sequencing of HD individuals associated with delayed HD onset. In fact, damaging PMS1 variants have been identified in both delayed and early onset HD individuals (McAllister et al 2022, NatNeurosci). Also, an association of a predicted damaging PMS1 variant with delayed HD onset does not suggest an association of PMS1 with CAG expansions. Please revise this sentence. Either way, the existence of damaging PMS1 variants with early and delayed HD onset complicates the ability to predict the effect of targeting PMS1.

11. That PMS1+/- shows no effect on CAG instability relative to PMS1+/+ or PMS1-/-, presents a conundrum. Might PMS1 transcript and/or protein levels may be maintained at fixed levels, regardless of gene-copy. For example, see (Leung et al, 2000, JBC; Trojan j et al, 2002, Gastroenterol; Cannavo et al 2005, CancerRes). This should be assessed on the PMS1+/-, PMS1+/+, and PMS1-/- RPE cell lines.

11a. That PMS1+/- shows no effect on CAG instability relative to PMS1+/+, is an unfortunate absence of a dose effect, at least at the DNA level, that could be supportive of a partial effect of a possible therapeutic. The inability to have a dose effect does not bode well for a possible anti-PMS1 therapeutic. Recent findings of 40-50% silencing of MSH3 with di-valent anti-Msh3 siRNAs slowed somatic expansions in HD mouse brains (O'Reilly et al, 2023, MolTherap). Based on McLeans findings, to slow expansions targeting PMS1 would have to be greater than 50% and approach 100%. Please comment.

12. The variable effects of risdiplam (reduced CAG expansions) relative to branaplam (increased CAG expansions) on cells with one PMS1 allele refractory to splice modulation, draws in question the ability to predictably target CAG instability. It is unknown what direction, expansions, or stabilization might arise. The authors suggest that this variability may be due to the off-target effects of the splice modulators on other HD modifiers, as it is known that they can affect many genes, including LIG. They should also mention FAN1, MLH1, MLH3, FANCI (Ottesen). These diverse effects reduce the ability to predictable the effect of PMS1 targeting upon CAG instability. Moreover, that fact that variants of CAG modifier genes may differentially affect their ability to be modulated by branaplam, risdiplam, or other splice modulators, as shown by these authors, adds further unknown levels of unpredictable variability.

Minor concerns:

13. The data of Fig. 4C and Supp Fig. 4B, x-axis, should be displayed as repeat number, not bp.

14. Line 201-2, the statement "We validated the relevance of our RPE1-AAVS1-CAG115 cell line to model somatic instability processes by perturbing modifiers of HD age-at-onset predicted to influence repeat instability" this is inappropriate logic, modifiers of AAO are not necessarily predicted to affect instability. Something more along the lines of "We validated the utility of the cell model by modulating known modifiers of CAG instability."

15. Line 229, the statement "Overall, these results are consistent with the effects of these HD genetic modifiers in HD individuals and animal and other cell models."

16. Line 361-3, the statement: "However, damaging PMS1 variants in exome sequencing of HD individuals associated with extremely delayed HD onset suggest that reduced PMS1 function suppresses somatic CAG expansion 31." The cited study shows no data of CAG instability for any PMS1 variants. Please modify the sentence.

17. Line 381: that reads "As an example, high risdiplam dosage results in downregulation of another HD genetic modifier, LIG1 33" , should read "As an example, high risdiplam but not branaplam dosage results in downregulation of another HD genetic modifier, LIG1 33. This differential effect of the two splice site modifiers might be a possible explanation for their differential effects upon CAG expansion."

18. Line 325-6: "...dose-dependent effect of branaplam on preventing CAG repeat expansion, which decreased 1.2-fold..." should read, "...dose-dependent effect of branaplam on reducing CAG repeat expansion, which decreased 1.2-fold..."

19. The Discussion should comment upon their findings of PMS1 on CAG stability, relative to published findings of MLH1, PMS2, and MLH3 in CAG instability in mammalian/human models. The data of Fig.4B for PMS1^{-/-}, while diminished relative to WT and PMS1^{+/-}, appears to be expanding by 3-4 repeats after 28 days in culture. Thus, effect is not all-or none and may be redundant with PMS2, and MLH3.

20. That PMS1^{+/-} cells show unaltered CAG instability profiles relative to controls suggests that, unlike MSH3^{+/-}, there is no dosage effect for PMS1 indicating that approaches to knock-down or inhibit PMS1 may need to be more severe than for MSH3. O'Reilly recently found that siRNA KD approaches were able to reduce MSH3 by as much as 40-50% with dramatic effect upon suppressing expansions. In vivo Pms1^{+/-} and Pms1^{-/-} HD mouse results are wanting.

21. Figure S11: Panel A is missing. Legend does not describe difference between panels B and C?

22. The text is prematurely forward-looking regarding the ability to make claims of PMS1 being a potential therapeutic target. Considering the unaddressed cancer safety concerns of targeting PMS1 in humans, the known off-target effects of splice modulators, and the known adverse effects (peripheral neuropathy) of branaplam, the absence of functional knowledge of PMS1, absence of in vivo evidence of PMS1 in CAG instability/disease, that damaging variants of PMS1 associate with both disease hastening and delaying, this reviewer suggests a more cautious approach.

Reviewer #3 (Remarks to the Author):

The study provides several intriguing details and insights (e.g., the influence of exon 49 final nucleotide on exon 50b and the use of ddPCR for quantifying mis-splicing). However, there are some significant limitations that need to be addressed.

(1) The authors have developed a nice system to validate the causal impact of DNA repair genes on repeat expansion in Huntington's disease (HD). However, it is already known from GWAS studies on HD that DNA repair genes play a role in repeat expansion. While the validation presented in this study is solid, it alone does not seem significant enough to justify publication in this journal.

(2) The claim that the repeat expansion-suppressing effect of branaplam and risdiplam is mediated by PMS1 pseudoexon inclusion is not clearly demonstrated. The authors failed to generate a homozygous deletion of the PMS1 pseudoexon 5' splice site and instead used suboptimal cells with heterozygous deletion. The results from this experiment are unexpected and confusing, particularly the higher repeat gain in the heterozygous cells compared to wild-type cells on branaplam treatment (Fig 7e). The authors' explanation for this finding is not convincing enough, and the results do not fully support their conclusion. It is recommended that the authors establish a cell line that lacks the PMS1 pseudoexon homozygously.

(3) The authors reference a paper (ref 18) that claims reducing huntingtin levels by ASO treatment also reduces repeat expansion and provides data supporting the suppressive effect of transcription on repeat expansion (Fig. 4a). The authors should provide clearer explanations regarding how these factors, along with DNA repair, can impact repeat expansion.

RESPONSE TO REVIEWER COMMENTS

The comments of all of the Reviewers have been addressed below and resulted in a much-improved presentation. Each reviewer comment or related set of comments is shown verbatim in italics with the response shown beneath it in normal font.

Line numbers referred to in the responses are the line numbers shown with “simple markup” in Microsoft Word, since “all markup” results in discontinuous line numbers.

Reviewer #1:

This manuscript presents an interesting finding about the role of splice modulation in modulation of both HTT mRNA levels as well as somatic expansion of mHTT CAG repeat length. The authors present a compelling case for the role of the splice modulator drugs risdiplam and branaplam in the regulation of inclusion of a pseudoexon in PMS1 in addition to their reported role in regulation of inclusion of a translational inhibiting pseudoexon in HTT as well as describing two variants of the drug-induced alternative splicing of HTT. Their investigation of variant alleles in proximity to the HTT pseudoexon and the impact these variants have on activity of the splice modulators further supports the observations of several groups regarding the breadth of splicing biology impacted by these molecules. As relates to PMS1 alternative splicing, the authors show – consistent with the cited reports of PMS1 variants regulating somatic CAG repeat expansion – that drug-induced alternative splicing of PMS1 can regulate the rate of CAG expansion in an in vitro model, suggesting that treatment of HD patients with splice modulator drugs may have salutary effects via both inhibition of HTT mRNA / protein expression and the degree of somatic mHTT CAG repeat expansion. Overall, the work is quite interesting and represents interesting developments around both the MoA of splice modulator drugs and validation of the MoA of modifiers of mHTT pathology. It would be interesting (but beyond the scope of this manuscript) to see this work expanded into models of mutant HTT either in vitro or in vivo to further validate the impact of PMS1 and alternative splicing in regulation of the CAG repeat expansion and disease pathology.

One area of the manuscript that seems misplaced, or not well connected to the overall ‘story’ is the section featuring a discussion of the genome-wide SpliceAI predictions and alternative splicing by branaplam. This is not well integrated with the role of splice modulator drugs on HTT and PMS1 alternative splicing and their impact on the function and potential pathogenesis of HD. In addition, the work – while advancing from the variant allele data from the HTT transcript to predict additional alternative splicing events induced by branaplam – seems to provide little novel information beyond the several studies reported on transcriptome-wide alternative splicing induced by these drugs. I would suggest either better integrating this section into the overall section regarding HTT variant alleles (and contrasting with the noted lack of variant alleles around PMS1), or reserving this data for an independent manuscript that explores the role of these variant alleles and potential on-mechanism / off-target consequences of treatment with splice modulator drugs.

Overall, I would recommend the publication – following addressing the above noted major change and some minor notes below.

We thank the reviewer for their many positive comments on our work. The reviewer raised one major point and several minor corrections (listed later below).

Major point:

The reviewer felt that the section concerning genome-wide SpliceAI predictions and drug-induced splice modulation was “*misplaced or not well connected to the overall story*”, “*seems to provide little novel information beyond the several studies reported on transcriptome-wide alternative splicing induced by these drugs*” and suggested either “*better integrating it*” or “*reserving the data for an independent manuscript*”.

Response:

We agree that the section is better placed elsewhere in the flow of the manuscript. In the revised manuscript we have moved it to the final section of the Results and moved the former Figure 3 to become the new expanded Figure 8. We decided in favor of inclusion and against reserving the data for an independent manuscript for three major reasons. First, Reviewer #2 had positive comments concerning this section and asked that it be expanded with further work. Second, we have added in the penultimate section of the Results in the revised manuscript and in Figure 8a an analysis showing the lack of extensive overlap in the results of the other studies testing transcriptome-wide alternative splicing induced by these drugs. None of these other studies performed direct comparisons with the others or considered the effects of genome-wide genetic variation, so the incorporation of the transcriptomic comparisons and SpliceAI analysis does present novel information germane to interpreting findings with splice modulators. Finally, and most importantly, we believe that it is crucial in a paper that uses splice modulators and discusses their potential utility as therapeutic agents to remind the reader that human genetic variation is an important consideration for both on-target and off-target effects in developing any DNA sequence-targeted therapy. Our concluding section of the Results makes that point for off-target effects with experimental data while providing an initial bioinformatic approach to predict such variable “off-target” complications. We believe that it is an important contribution of this work that belongs in this paper.

Minor corrections:

Line 46 – sentence should start ‘These features’ (assuming this is referring to ‘some DNA repair genes’)

This typo was corrected – now line 48

Line 72 – suggest including ‘pharmacodynamic’ in the sentence ‘We found that the (pharmacodynamic) effectiveness of...’

This suggestion was implemented – now line 75

Line 273 – should include ‘to’ in the sentence ...figure 8b), predicted (to) result in ...

This typo was corrected – now line 300

Line 341 – FDA is the ‘Food & Drug Administration’

This error was corrected --- now line 517-518

Figure 2A – suggest including a box – line figure showing HTT exons 49 and 50 (and the pseudoexon) to better highlight the variants and where they are relative to the exons

This helpful suggestion was implemented in Figure 2a

Reviewer #2

McLean et al address a topic that is of great interest for biology, potential and ongoing therapeutics, and is timely. They report some of the first data of and effect of PMS1 on CAG instability, as well as a complex effect of two splice modulators, branaplam and risdiplam, upon CAG instability. This study broadens the awareness of the many genes that can be targeted by these splice modulators, and the downstream effects that they may have. They identify variants of HTT that affect the efficacy of the splice modulators. They present a predictor to identify such splice site variants – properly developed this could be of great utility to all splice-modulation approaches. They begin to extend the action of branaplam and risdiplam to PMS1 splice modulation. One of the more exciting advances of this study is the beginnings of the establishment of a new model for CAG repeat instability. The cell shows an effect of transcription across the repeat, albeit the reverse of what is expected. The cell shows the expected suppressed and enhanced CAG expansions by knock-outs of MSH3 and FAN1, respectively. This study provides the first data supporting a role of PMS1 in enhancing CAG expansions. While the effect of PMS1 is milder than MSH3, the finding has implications for the biology of CAG instability and may have implications for therapeutics. Interestingly, the authors tested whether branaplam and risdiplam treatment affected CAG instability. This was based upon the suggestive claims of a bioRxiv submission of >3 years ago (their ref 18), of an unsubstantiated finding that lowering of HTT protein affects CAG instability, the authors tested whether branaplam and risdiplam treatment might affect somatic CAG instability. They reveal that branaplam treatment can slow CAG expansion rates in the cell model. They rule-out an effect of HTT splice modification in splice-modulator altered CAG expansions, and provide some support for a role of PMS1 splice modification in reducing CAG expansions.

While there are some potentially exciting novel findings in this study, there are numerous serious concerns outlined below:

We thank the reviewer for their very careful reading of the manuscript, positive comments and very helpful criticisms and suggestions. The review indicated to us that we were not clear in conveying the intended messages in the paper and so we have added data and rewritten a number of sections for better orientation and clarity.

Major concerns:

The cell line is an exciting advance, showing biased expansions over short culture times and expected increased and reduced expansions by knockouts of FAN1 and MSH3, respectively. Careful analysis and awareness of the model system will support the utility and relevance of the RPE1-AAVS1-CAG115 cell model as “a useful model for functional genomic investigations of CAG repeat instability”, as claimed.

A number of concerns (below) centered on the benefits of better characterizing our cell model, so we added a section to the Results (beginning line 157 – A novel cell culture model for investigating CAG repeat instability) describing further work to characterize the model.

We have performed further characterization and analyses as raised in the following points made by the reviewer:

1. *The new model of CAG instability requires detailed characterization for its reflection of what is occurring in HD patient cells/tissues, or tissues of HD mouse models. For example, i) what is*

the endogenous AAVS1-driven direction of transcription across the CAG/CTG repeat? Is transcription direction the same as in the endogenous HTT gene? A schematic figure with promoters, repeat strands, and primers, would be helpful; ii) What is the inducible direction of transcription? Again, a schematic would be helpful; iii) Does the absence of transcriptional (doxycycline) induction result in an absence of transgene transcription, or only a reduced level? 1a. While not necessary for cell model establishment, it would interesting to know viii) Are the effects of each tested modifier (MSH3, FAN1, PMS1) dependent upon transcriptional alteration across the expanded repeat? ix) Do the modifiers require absence of cell proliferation to alter instability?

We have expanded the characterization of the cell line model in a section of the Results (lines 157-256). This includes insertion of the requested schematic figure (Figure 3a) delineating the transgene features and direction of transcription relative to the host *PPP1R12C* gene. We have also performed the requested quantification, showing that there is considerable transcription of the *HTT* CAG repeat in the absence of inducer (Figure 3g), supporting its potential involvement in instability of the CAG repeat. In the uninduced state there is also transcription in the opposite direction across the *HTT* exon 1 CAG repeat as part of host gene expression. Treatment with inducer increases expression of the *HTT* exon 1 transgene but reduces transcription in the opposite direction (Figure 3h). Induced cells still show CAG repeat instability, but less than the uninduced cells (Figure 3e&f), suggesting that the balance of bidirectional transcription influences the rate of instability, as has been suggested previously for such convergent transcription.

iv) Is instability affected by the absence of cell proliferation? v) The G0/1 arrest of cell proliferation under the test conditions used must be demonstrated; vi) If claims are to be made or inferred, as it is, that instability, in such small, 96-well cultures, arise during G0/1, the amount of cell proliferation following seeding to G0/1 arrest should be determined. vii) It appears that there is proliferation, since the description of the results in Fig. 5 state that the cells were maintained at confluency, but increasing doses of branaplam decreased cell proliferation. this can be of concern if instability in the cell model occurred during proliferation.

Dividing cells show a higher rate of CAG expansion compared to confluent cells, but by only ~1.9-fold (new Figure 3f). In the 96-well cultures, confluence is reached within 4 days (Figure 3 b, c) and CAG repeat instability continues throughout the course of a multi-week experiment (Figure 3f).

We apologize for the confusion raised by our previous presentation of Figure 5 and Supplementary Figure 5. We had performed experiments with branaplam both at confluency (Figure 5) and in dividing cultures (Supplementary Figure 5b) but were not sufficiently clear in distinguishing these experiments in the text. We have edited the text to clarify (lines 271-276).

2. While the AAVS1 locus has been suggested as a safe harbour since it is permissive for stable transgene expression, recent evidence reveals that considerable inter-clone variability of AAVS1 knock-ins does occur, as does transgene silencing, demanding careful attention to transgene expression (Bhagwan JR et al, 2019, F1000Res, 8:1911). Since transcription across the expanded repeat is well-established as a major driver of CAG instability, at a minimum it is crucial that levels of the transgene transcript be quantified under the various test conditions used.

In the revised manuscript, we have used two methods to assess this question.

First, we measured productive expression of the transgene by the EGFP signal produced. This demonstrated considerable expression of the transgene in the uninduced state and increased expression in the induced state (Figure 3g). A minor population of EGFP negative cells was present in both circumstances. We separated the EGFP-negative and EGFP-positive cells and found that after a short time in culture the majority of the originally EGFP-negative cells had become EGFP-positive (Supplementary Figure 3c) indicating that the GFP-negativity represents a transient state rather than permanent transgene silencing. We also tested both populations and found no difference between them in CAG repeat instability in either the uninduced or induced state (Supplementary figure 3d). Furthermore, we note that the CAG repeat distribution remains a single normally distributed population as it expands, suggesting a relatively homogenous population. In a hypothetical scenario where permanent transgene silencing affected CAG repeat instability, we would expect a bimodal distribution with two populations with different rates of CAG repeat instability (e.g., the distribution observed with *MSH3* knock-out in Figure 4a).

Second, we used digital droplet PCR (ddPCR) with probes to assess the *HTT* exon 1 transgene, transcribed in one direction, and the puromycin resistance gene and host *PPP1R12C* gene, transcribed in the opposite direction. For the latter, a heterozygous SNP in exon 2 enabled us to distinguish between formation of the spliced exon1-2 products from the transgene-containing and wild-type chromosomes in our cell model with heterozygous transgene integration. The results in Figure 3h show that *HTT* exon 1 transgene expression is relatively higher than the other genes in the uninduced state and the difference is greatly accentuated in induced cells.

3. Oddly, under non-induced/no-transcription across the repeat of the transgene, showed greater CAG expansions compared to induced conditions (Fig 4A). This is in opposition to many published findings where transcription across the expanded repeat enhances repeat instability (this is true in many model systems, many organisms, and HD mice, at the level of pol II elongation) and flies in the face of the ubiquitously expressed WT and mutant HTT in patients brains. This is very strange and should be addressed relative to published findings and relevance. This concern strengthens the need to characterize the models as outlined above (points 1 & 2). The authors may wish to revisit the Goula et al studies and interpretations (Goula et al, 2012, PLoS Genet 8:e1003051; Goula et al, 2013, Transcription, 4: 172-6).

We agree that this result was unexpected, but the finding is consistent across all of our studies. However, since there is considerable productive transcription across the *HTT* exon 1 in the uninduced state, we don't believe that our results argue with the need for such transcription to support CAG expansion. Instead, since the increased transcription that results from induction is accompanied by reduced transcription in the opposite direction, our data may reflect the stimulating effect of convergent transcription on CAG repeat instability (e.g., Nakamori, et al. HMG 2011, Lin et al. MCB 2010).

3a. For the induced versus non-induced conditions, raw GeneMapper profiles should be provided (multiple clonal lines, not just pooled cells), rather than the batch processed data of Fig 4A.

We have provided these data as Supplementary Figure 3b

4. The data that rule-out a role of drug-induced CAG instability through HTT splicing (and lowered HTT protein) is strong. Impressive, crafty, and clean experimentation. Nice! The authors should comment on this finding relative to the suggestion that lowering HTT protein modulates

somatic CAG expansions (Coffey et al, their citation #18). It is possible that the effect observed by Coffey et al is likely due to ASO-induced transcriptional arrest at the HTT gene, as has been observed (Lee & Mendell, 2020, *MolCell*, 77:1044-1054; Lai F et al., 2020, *MolCell*, 77:1032-1043; Nakamori M et al., 2011, *MolTher*, 19:2222-7) and shown to reduce somatic repeat instability (Nakamori M et al., 2011, *MolTher*, 19:2222-7). Similarly, recent evidence showing that the near-complete silencing of HTT with di-valent anti-HTT siRNAs had no measurable effect on somatic repeat expansions in HD mouse brains (O'Reilly et al, 2023, *MolTherap*).

We have now more explicitly stated that our results differ from the suggestion by Coffey et al. and we have presented the suggestion of ASO-induced transcriptional arrest as a possible explanation (lines 343-349).

5. In contrast, the data supports some ability to connect, in part, the risdiplam-induced reductions in CAG expansions to altered PMS1 splicing, *LIG1*, *FAN1*, *MLH1*, *MLH3*, or any of the many DNA repair or other genes altered by their treatment (Bhattacharyya et al 2021 *NatComm*; Ottesen et al 2023 *NAR*). Is not possible without further experiments to be anything but suggestively connected to PMS1 altered splicing.

5a. Because the published effects of branaplam and risdiplam on *LIG1*, *FAN1*, *MLH1*, *MLH3*, and many DNA repair genes, but not of PMS1 (Bhattacharyya et al 2021 *NatComm*; Ottesen et al 2023 *NAR*) it is critical that the authors acknowledge that the authors acknowledge that the effects may vary between cell lines/types. For example those studies also did not find any effect of those drugs on the levels of PMS1, however, McLean and colleagues do find PMS1 differences in their branaplam-treated LCLs. Thus other genes not affected in one cell line, may be affected in another. Moreover, McLean et al did not assess the effect of branaplam or risdiplam on PMS1 levels in the RPE1-AAVS1-CAG115 cell line they used for CAG instability. At a minimum, the authors must assess these.

5b. The effect of branaploam and risdiplam upon PMS1 splicing in LCLs is convincing. This must be done in the RPE cells under test conditions, and must be followed-up with evidence for transcript and PMS1 protein reduction. A western blot is a must.

5c. Line 331-6: "Overall, the results of targeting PMS1 via the drug inducible pseudoexon explained the reduction in rate of CAG repeat expansion caused by branaplam but only partially explained the observed effect with risdiplam. The partial effect with the latter along with the increases in expansion with the lower branaplam doses, suggest that the drugs may also have effects on splicing in other genes that influence CAG repeat instability." These authors cannot conclude that the branaplam-induced alterations in CAG expansions are explained by targeted PMS1 splicing. Both branaplam and risdiplam targeting of PMS1 splicing can only partially explain the altered CAG expansion rates induced by these drugs. The data do suggest that the effect of branaplam more directly involves PMS1 than does risdiplam.

Our explanation of the experiments with the heterozygously-edited PMS1 pseudoexon cells was poorly presented and led to confusion. We have now rewritten this section more clearly to indicate why these experiments support the conclusion that branaplam and risdiplam contribute to reducing the rate of CAG expansion via PMS1 pseudoexon inclusion (Lines 350-386). We have further emphasized explicitly, as noted by this reviewer, that the contribution of risdiplam does not fully account for the observed reductions, suggesting that there are other genes that

are contributing beyond *PMS1* (We have also elaborated on both of these points below in response to Reviewer #3). Finally, we have explored the datasets from other studies (including the Ottesen et al 2023 NAR paper that found more than 10,000 genes whose expression was altered by risdiplam). These data do not point to a clear candidate to explain the residual effects after accounting for the *PMS1* pseudoexon effect. However, this addition (Lines 387-407, Figure 8a) emphasizes the complexity of these drugs' genome-wide effects, including tissue-specific differences in gene targets, and is an important caution for their use as therapeutic agents.

Moreover, McLean et al did not assess the effect of branaplam or risdiplam on PMS1 levels in the RPE1-AAVS1-CAG115 cell line they used for CAG instability. At a minimum, the authors must assess these. The effect of branaplam and risdiplam upon PMS1 splicing in LCLs is convincing. This must be done in the RPE cells under test conditions, and must be followed-up with evidence for transcript and PMS1 protein reduction. A western blot is a must.

We have now complemented the LCL analysis with western blot analysis (Figure 6c) showing dose-dependent reduction of *PMS1* in our RPE1-AAVS1-CAG115 cell line with both branaplam and risdiplam, with controls of untreated unedited cells and biallelically edited *PMS1* knock-out cells.

6. The HD clinical trial of branaplam was halted due to concerns of safety and off-target splice modulation. Recent evidence for branaplam and risdiplam reveals that “both compounds triggered massive perturbations of splicing events, inducing off-target exon inclusion, exon skipping, intron retention, intron removal and alternative splice site usage” (Bhattacharyya et al 2021 NatComm; Ottesen et al 2023 NAR). Notably risdiplam was reported to downregulate a massive number of genes, including those that are known to affect CAG instability (MLH1, MLH3, FAN1, LIG1, and FANCI) cell cycle regulation, DNA replication, base excision repair, homologous recombination, and more (Bhattacharyya et al 2021 NatComm; Ottesen et al 2023 NAR). An analysis of the effects of branaplam and risdiplam treatments upon protein levels of MLH1, MLH3, PMS2, FAN1, LIG1, and PMS1 would minimize concerns of off-target effects over PMS1, in their role in splice-modulator induced CAG instability.

We agree that the splice modulators present many concerns when contemplated as therapeutic agents. The message of our paper is to support *PMS1* as a potential therapeutic target for HD, not to suggest that this therapeutic action necessarily be accomplished by splice modulation. The splice modulators resulted in our discovery and assessment of the pseudoexon, but our CRISPR-Cas9 knock-out of *PMS1* in a tractable cell model directly supports its effect on the rate of CAG expansion. This message was poorly elaborated, and our presentation led to confusion concerning this issue. Consequently, we have extensively rewritten the text to clarify this critical point, including modifying the title, abstract and discussion to make it clear that our data support *PMS1* as a target, that the splice modulators that led us to these experiments represent only one possible route to address the target, but that they raise many considerations to take into account, including tissue-specific off-target effects and effect of genetic variation if such drugs are to be used in patients. Further exploration of the drugs' residual effects on CAG repeat instability after accounting for the contribution of *PMS1* will be very difficult given the complexity of the drug response, though it might prove fruitful and interesting. However, it is beyond the scope of the current study.

6a. The concerns of the off-target effects of the modulators demands an ability to predict their targets. Excitingly, McLean et al have developed this prediction using SpliceAI. The data in

Figure 3 are predictions of branaplam, but not risdiplam, target sites based upon some, but not all of the published transcriptome-wide data. Specifically, the authors have selectively used the branaplam data from Monteys et al. 2021; Bhattacharyya et al., 2021; Keller et al.) and applied these to SpliceAI. They need to include the most current dataset of Ottesen 2023 (NAR), a study that they cite (their ref #33) for LIG1, which, amongst all the other hits of that study, was not included in the SpliceAI learning set. The branaplam data of the Ottesen study should be included.

6b. The authors make claims for both branaplam and risdiplam, but only use the published branaplam transcriptome-wide data for predictions and have neglected to apply the parallel risdiplam data for SpliceAI predictions. This should be done.

We have now included all branaplam and risdiplam data, including that from the Ottesen study in updated SpliceAI analyses and presentation in Figure 8.

7. Contractions are under-whelming. Regarding claims of contractions for MSH3^{-/-}, the strength of this claim should be supported by continuous contractions in multiple clonal lines over longer term cultures. Contractions are not evident in the GeneMapper profile of the clonal MSH3^{-/-} line (Fig. S4B). An average repeat loss of 0.037 repeat units/week derived from 11 clones 22 cultures, for 0 and 28 days, equates to an average loss of 1 repeat over 27 weeks, or a loss of 2 repeats over just more than a year. Seems that contraction events were likely evident in only a handful of clones/cultures, as is evident by the distributions (Fig. 4E). If true, this is very very low levels. What appears as contractions in pooled cells (Fig. 4C) is, as the authors suggest, likely due to a mixture of edited and non-edited cells. But could also be due to culture over-take by cells with shorter repeats. In vivo data in HD mice did not reveal repeat contractions in any tissue, but might have missed these in the short life of a mouse.

We agree and did not intend to claim that contractions were a meaningful result of the experiment. We have reworded this section to avoid placing undue significance on the very small changes observed (lines 235-236).

8. The limited size and rate of expansions in the PMS1^{-/-} lines is convincing, as they seem to be biased for each line (Fig. 4E) and are readily evident in the clonal line (Fig. S4B). Data of Fig. S4B should have dots aligning the upper and lower repeat profiles, as done in Fig. 4C. This would facilitate an appreciation of the limited expansions.

We have modified Supplementary Figure 4b as suggested.

9. Line 356: the statement “Additionally, in contrast to Mlh1 and Pms2, the loss Pms1 does not cause tumors in mice 26.”, should be specific, “Additionally, in contrast to Mlh1 and Pms2, the loss Pms1 does not cause tumors in mice, aged to only 12-months 26.” Confidence of safety should be toned-down. Some mention of the published known effects of human cancers and PMS1 or MSH3 heterozygosity, LOH, and homozygous mutations (somatic and germline), CMMRD, must be covered. For example, see (Hamad & Ibrahim 2022, HeredCancerClinPract, 20:16; Alghamdi et al, 2023, JEndocrSoc, 7:bvad035; Wang et al, 2013, Gene, 524:28-34). Clearly, while germline and somatic mutations in PMS1 may be rare in humans, they are not without serious life-impacting effects. Targeted reduction or inhibition of PMS1 for life-long administration, as would be needed for HD, could well have deleterious effects.

We agree. The impression we conveyed of confidence in the safety of targeting PMS1 was not intended and we have rewritten the Discussion to include these issues (lines 475-490).

10. Line 361-: *The authors have selectively mentioned only that damaging PMS1 variants in exome sequencing of HD individuals associated with delayed HD onset. In fact, damaging PMS1 variants have been identified in both delayed and early onset HD individuals (McAllister et al 2022, NatNeurosci). Also, an association of a predicted damaging PMS1 variant with delayed HD onset does not suggest an association of PMS1 with CAG expansions. Please revise this sentence. Either way, the existence of damaging PMS1 variants with early and delayed HD onset complicates the ability to predict the effect of targeting PMS1.*

This statement has been reworded for accuracy (lines 463-466)

11. *That PMS1+/- shows no effect on CAG instability relative to PMS1+/+ or PMS1-/-, presents a conundrum. Might PMS1 transcript and/or protein levels may be maintained at fixed levels, regardless of gene-copy. For example, see (Leung et al, 2000, JBC; Trojan j et al, 2002, Gastroenterol; Cannavo et al 2005, CancerRes). This should be assessed on the PMS1+/-, PMS1+/+, and PMS1-/- RPE cell lines.*

We have now demonstrated by western blot that PMS1 protein levels in the heterozygous cells are comparable to wild-type levels (Supplementary Figure 4c), as the Reviewer suggested might be the case. We have also suggested the potential importance of binding partner stoichiometry in regulating the PMS1 level (lines 247-251 and lines 493-495).

11a. *That PMS1+/- shows no effect on CAG instability relative to PMS1+/+, is an unfortunate absence of a dose effect, at least at the DNA level, that could be supportive of a partial effect of a possible therapeutic. The inability to have a dose effect does not bode well for a possible anti-PMS1 therapeutic. Recent findings of 40-50% silencing of MSH3 with di-valent anti-Msh3 siRNAs slowed somatic expansions in HD mouse brains (O'Reilly et al, 2023, MolTherap). Based on McLeans findings, to slow expansions targeting PMS1 would have to be greater than 50% and approach 100%. Please comment.*

We have now included this comparison and its potential ramifications in the Discussion (lines 491-503).

12. *The variable effects of risdiplam (reduced CAG expansions) relative to branaplam (increased CAG expansions) on cells with one PMS1 allele refractory to splice modulation, draws in question the ability to predictably target CAG instability. It is unknown what direction, expansions, or stabilization might arise. The authors suggest that this variability may be due to the off-target effects of the splice modulators on other HD modifiers, as it is known that they can affect many genes, including LIG. They should also mention FAN1, MLH1, MLH3, FANCI (Ottesen). These diverse effects reduce the ability to predictable the effect of PMS1 targeting upon CAG instability. Moreover, that fact that variants of CAG modifier genes may differentially affect their ability to be modulated by branaplam, risdiplam, or other splice modulators, as shown by these authors, adds further unknown levels of unpredictable variability.*

The complexity pointed out by the reviewer has now been addressed in comparing the data from previous studies in the Results (lines 387-407), including effects of the drugs on those other genes known from human genetic studies to modify HD. However, while this complexity affects targeting by splice modulation, it does not similarly complicate the predictable targeting of CAG instability via PMS1 using other types of interventions (e.g., a CRISPR-Cas9 strategy like the one used here to create a *PMS1* knock-out). We have rewritten the Discussion to convey that message – that our data support *PMS1* as a target for downregulation to reduce

CAG repeat instability regardless of what method is used to achieve that downregulation (lines 471-474). Whether that targeting leads to a useful treatment, given the potential delivery and safety considerations remains to be determined.

Minor concerns:

13. *The data of Fig. 4C and Supp Fig. 4B, x-axis, should be displayed as repeat number, not bp.*

Both Figure 4c and Supplementary figure 4b have been modified as indicated.

14. *Line 201-2, the statement “We validated the relevance of our RPE1-AAVS1-CAG115 cell line to model somatic instability processes by perturbing modifiers of HD age-at-onset predicted to influence repeat instability” this is inappropriate logic, modifiers of AAO are not necessarily predicted to affect instability. Something more along the lines of “We validated the utility of the cell model by modulating known modifiers of CAG instability.”*

We agree, have reworded for accuracy (lines 361-364). We have also made it clear in the Discussion that our data are consistent with *PMS1* being both the source of HD modification by a GWAS locus on chromosome 2 and a contributor to the rate of CAG expansion in our cell line (lines 448-474)

15. *Line 229, the statement “Overall, these results are consistent with the effects of these HD genetic modifiers in HD individuals and animal and other cell models.”*

We have reworded this statement for accuracy (now lines 252-254)

16. *Line 361-3, the statement: “However, damaging *PMS1* variants in exome sequencing of HD individuals associated with extremely delayed HD onset suggest that reduced *PMS1* function suppresses somatic CAG expansion 31.” The cited study shows no data of CAG instability for any *PMS1* variants. Please modify the sentence.*

We have reworded this statement for accuracy (now lines 463-466)

17. *Line 381: that reads “As an example, high risdiplam dosage results in downregulation of another HD genetic modifier, *LIG1 33*”, should read “As an example, high risdiplam but not branaplam dosage results in downregulation of another HD genetic modifier, *LIG1 33*. This differential effect of the two splice site modifiers might be a possible explanation for their differential effects upon CAG expansion.”*

We have expanded this point to clarify that there are effects of the drugs on many other genes and have delineated the effects on those genes that are also known to be modifiers of HD. However, while we have demonstrated an effect of the *PMS1* pseudoexon inclusion, the residual effects of risdiplam (and branaplam) not accounted for by *PMS1* are not readily attributable from these data to any evident candidate gene (Lines 387-407) and a deeper experimental dive into this question is beyond the scope of the present study.

18. *Line 325-6: “...dose-dependent effect of branaplam on preventing CAG repeat expansion, which decreased 1.2-fold...” should read, “...dose-dependent effect of branaplam on reducing CAG repeat expansion, which decreased 1.2-fold...”*

We have corrected this wording. (now lines 371-372)

19. *The Discussion should comment upon their findings of PMS1 on CAG stability, relative to published findings of MLH1, PMS2, and MLH3 in CAG instability in mammalian/human models. The data of Fig.4B for PMS1^{-/-}, while diminished relative to WT and PMS1^{+/-}, appears to be expanding by 3-4 repeats after 28 days in culture. Thus, effect is not all-or none and may be redundant with PMS2, and MLH3.*

Figure 4b actually reflects a much lower rate of CAG expansion of 0.13/wk or ~ 0.5 repeats after 28 days in culture. We have commented on the possibility that this low level reflects partial redundancy in the mismatch repair machinery. (lines 240-242 and 509-512)

20. *That PMS1^{+/-} cells show unaltered CAG instability profiles relative to controls suggests that, unlike MSH3^{+/-}, there is no dosage effect for PMS1 indicating that approaches to knock-down or inhibit PMS1 may need be more severe than for MSH3. O'Reilly recently found that siRNA KD approaches were able to reduce MSH3 by as much as 40-50% with dramatic effect upon suppressing expansions. In vivo Pms1^{+/-} and Pms1^{-/-} HD mouse results are wanting.*

This important difference has now been addressed in the Discussion (lines 491-503)

21. *Figure S11: Panel A is missing. Legend does not describe difference between panels B and C?*

This Figure and legend have been corrected

22. *The text is prematurely forward-looking regarding the ability to make claims of PMS1 being a potential therapeutic target. Considering the unaddressed cancer safety concerns of targeting PMS1 in humans, the known off-target effects of splice modulators, and the known adverse effects (peripheral neuropathy) of branaplam, the absence of functional knowledge of PMS1, absence of in vivo evidence of PMS1 in CAG instability/disease, that damaging variants of PMS1 associate with both disease hastening and delaying, this reviewer suggests a more cautious approach.*

We agree and had not intended to minimize safety concerns. We have modified the text throughout to convey our message more clearly. The safety message is summarized in a rewritten Discussion that outlines the issues that must be considered in addressing PMS1 as a target for downregulation, regardless of how that downregulation is achieved, as well as the issues involved in using splice modulators to achieve the goal.

Reviewer #3

The study provides several intriguing details and insights (e.g., the influence of exon 49 final nucleotide on exon 50b and the use of ddPCR for quantifying mis-splicing). However, there are some significant limitations that need to be addressed.

(1) The authors have developed a nice system to validate the causal impact of DNA repair genes on repeat expansion in Huntington's disease (HD). However, it is already known from GWAS studies on HD that DNA repair genes play a role in repeat expansion. While the validation presented in this study is solid, it alone does not seem significant enough to justify publication in this journal.

As we have clarified in the Discussion of our revised manuscript, *PMS1* has not previously been experimentally demonstrated to influence CAG repeat instability. It lies near a genome-wide significant GWAS signal for modification of HD, but has not been demonstrated to be the source of that signal. Moreover, while other mismatch repair genes have been demonstrated to modify both HD and CAG repeat instability and *PMS1* is classified as a DNA mismatch repair gene, its role in canonical mismatch repair remains unclear. Consequently, we believe that demonstrating that *PMS1* can influence CAG repeat instability and is therefore a valid target for downregulation as a potential treatment for HD and other CAG repeat disorders is significant. We also believe that our findings with respect to the splice modulators represent valuable insights for those aiming to use these molecules as therapeutic agents. Finally, we believe that the cell line that we report and have now more extensively characterized fills a major gap in the HD research armamentarium by permitting experimental investigation of CAG repeat expansion and testing of methods to interfere with it to be carried out in experiments lasting days to weeks rather than months to years.

(2) The claim that the repeat expansion-suppressing effect of branaplam and risdiplam is mediated by PMS1 pseudoexon inclusion is not clearly demonstrated. The authors failed to generate a homozygous deletion of the PMS1 pseudoexon 5'ss and instead used suboptimal cells with heterozygous deletion. The results from this experiment are unexpected and confusing, particularly the higher repeat gain in the heterozygous cells compared to wild-type cells on branaplam treatment (Fig 7e). The authors' explanation for this finding is not convincing enough, and the results do not fully support their conclusion. It is recommended that the authors establish a cell line that lacks the PMS1 pseudoexon homozygously.

We respectfully disagree that “The claim that the repeat expansion-suppressing effect of branaplam and risdiplam is mediated by *PMS1* pseudoexon inclusion is not clearly demonstrated.”

We believe that our data clearly demonstrate that *PMS1* pseudoexon inclusion influences CAG repeat expansion, but that we explained these data poorly and made their interpretation more difficult by our presentation.

We have tried to correct this in the revised manuscript with a step-wise delineation of our argument (lines 350-386) which we relate here:

“In contrast with *HTT*, we were unable to obtain a line homozygous for the removal of the *PMS1* pseudoexon, but this did not prevent a test of the role of splice modulator-induced pseudoexon inclusion. In the heterozygous lines, the removal of the *PMS1* pseudoexon from one allele

makes that edited allele refractory to pseudoexon inclusion, while the wild-type allele remains susceptible to the effect. Notably, our previous editing experiment to create *PMS1* knock-outs showed that a single active *PMS1* allele is sufficient to support CAG instability comparable to wild-type. Consequently, splice modulator treatment of the lines with heterozygous removal of the pseudoexon presented two possibilities: 1) if *PMS1* pseudoexon inclusion plays no role in reducing CAG repeat instability, then splice modulator treatment of these lines should show reduced CAG expansion comparable to treated wild-type cells, versus 2) if *PMS1* pseudoexon inclusion contributes to reducing CAG repeat instability, then splice modulator treatment should result in a restoration of CAG expansion comparable to untreated wild-type or heterozygous *PMS1*-knockout cells. The results of treating the cell lines with the splice modulators were consistent with the second alternative, supporting a contribution of pseudoexon inclusion to the reduced CAG expansion observed in treated wild-type cells. They also hinted at potential additional effects of the splice modulators beyond those accounted for by *PMS1*.”

We altered the data presentation in Figure 7c and 7d in order to make it easier for the reader to visualize the results in the context of the two alternatives delineated above. The data show that the presence of a single *PMS1* allele lacking a viable pseudoexon is sufficient to prevent branaplam’s ability to reduce CAG expansion, consistent with alternative #2 but not alternative #1. The fact that the rate of CAG expansion increases slightly above the wild-type level does not argue against pseudoexon inclusion being the mediator of branaplam’s ability to reduce CAG expansion in wild-type cells. Instead, it argues that once the contribution of the *PMS1* pseudoexon inclusion is eliminated by editing one allele, additional less robust effects of branaplam are detectable. These likely result from the action of branaplam at other genes, as we have suggested.

The same general argument applies to risdiplam. The data show that the presence of a single *PMS1* allele lacking a viable pseudoexon is sufficient to influence risdiplam’s ability to reduce CAG expansion, consistent with alternative #2 but not alternative #1. The fact that the rate of CAG expansion is not fully restored to the wild-type level does not argue against pseudoexon inclusion being a mediator of risdiplam’s effect on reducing CAG expansion in wild-type cells. Instead, it argues that *PMS1* pseudoexon inclusion contributes to risdiplam-induced reduction of CAG expansion but does not account for the entirety of risdiplam’s effect on reducing CAG expansion. The participation of other risdiplam-induced factors on reducing CAG repeat expansion, presumably by the action of risdiplam on other genes, becomes evident only when the ability of risdiplam to contribute via *PMS1* is eliminated by the pseudoexon editing.

Thus, we believe that the data presented fully support the conclusion that both branaplam and risdiplam can contribute to reducing the rate of CAG expansion by promoting *PMS1* pseudoexon inclusion and that once this effect of each drug is accounted for, additional drug effects that either promote (branaplam) or reduce (risdiplam) CAG repeat expansion become evident.

That said, we did try again to obtain a cell line homozygous for pseudoexon editing, but have so far been unable to do so. However, we do not agree that such a cell line is critical to this study since the heterozygous line fully enabled testing of the issue at hand. A homozygous line might in the future facilitate the identification, among the complex effects of branaplam and risdiplam, of what other genes are responsible for the non-*PMS1* contributions, but that is beyond the scope of the current study.

(3) The authors reference a paper (ref 18) that claims reducing huntingtin levels by ASO treatment also reduces repeat expansion and provides data supporting the suppressive effect of transcription on repeat expansion (Fig. 4a). The authors should provide clearer explanations regarding how these factors, along with DNA repair, can impact repeat expansion.

The disagreement of our findings with the Coffey et al preprint (ref 17) has now been clearly stated and potential explanations pointed out (lines 343-349).

The role of transcription has been extensively discussed as part of the more detailed characterization of the cell model (see added Results section – A novel cell culture model for investigating CAG repeat instability and our responses to Reviewer #2).

REVIEWERS' COMMENTS

Reviewer #1 (Remarks to the Author):

The authors have addressed the questions and comments raised in a largely appropriate manner and the article is significantly improved. The manuscript should be accepted for publication.

Reviewer #2 (Remarks to the Author):

The authors have satisfactorily addressed the concerns. This interesting study should be published.

Reviewer #2 (Remarks on code availability):

This reviewer is not an expert in code assessment and presumes this to be acceptable.

Reviewer #3 (Remarks to the Author):

While it is unfortunate the homozygous PMS1-pseudoexon-edited cell line could not be obtained, the authors have provided much clearer interpretation of their results in the revised manuscript. Additional explanations on other previously confusing observations, such as why ASO HTT knockdown and dox-induced forced expression of repeats both suppress repeat expansion, have helped clarify the seemingly conflicting observations. Overall, I commend the authors on their extensive and rigorous work. This manuscript is with a lot of insights, and I believe it will make a valuable contribution to the field.

RESPONSE TO REVIEWER COMMENTS

Each reviewer comment or related set of comments is shown verbatim in italics with the response shown beneath it in normal font.

Reviewer #1:

The authors have addressed the questions and comments raised in a largely appropriate manner and the article is significantly improved. The manuscript should be accepted for publication.

Response:

We thank the reviewer for their time and positive comments.

Reviewer #2 (Remarks to the Author):

The authors have satisfactorily addressed the concerns. This interesting study should be published.

Reviewer #2 (Remarks on code availability):

This reviewer is not an expert in code assessment and presumes this to be acceptable.

Response:

We thank the reviewer for their time and positive comments.

Reviewer #3:

While it is unfortunate the homozygous PMS1-pseudoexon-edited cell line could not be obtained, the authors have provided much clearer interpretation of their results in the revised manuscript. Additional explanations on other previously confusing observations, such as why ASO HTT knockdown and dox-induced forced expression of repeats both suppress repeat expansion, have helped clarify the seemingly conflicting observations. Overall, I commend the authors on their extensive and rigorous work. This manuscript is with a lot of insights, and I believe it will make a valuable contribution to the field.

Response:

We thank the reviewer for their time and positive comments.